# Feedback Guidance of Diffusion Models

Felix Koulischer[1,*]        Florian Handke[2]        Johannes Deleu[1]
Thomas Demeester[1,†]        Luca Ambrogioni[2,†]

[1] Ghent University - imec
[2] Donders Institute for Brain Cognition and Behaviour, Radboud University

## Abstract

While Classifier-Free Guidance (CFG) has become standard for improving sample fidelity in conditional diffusion models, it can harm diversity and induce memorization by applying constant guidance regardless of whether a particular sample needs correction. We propose **F**eed**B**ack **G**uidance (FBG), which uses a state-dependent coefficient to self-regulate guidance amounts based on need. Our approach is derived from first principles by assuming the learned conditional distribution is linearly corrupted by the unconditional distribution, contrasting with CFG's implicit multiplicative assumption. Our scheme relies on feedback of its own predictions about the conditional signal informativeness to adapt guidance dynamically during inference, challenging the view of guidance as a fixed hyperparameter. The approach is benchmarked on ImageNet512x512, where it significantly outperforms Classifier-Free Guidance and is competitive to Limited Interval Guidance (LIG) while benefitting from a strong mathematical framework. On Text-To-Image generation, we demonstrate that, as anticipated, our approach automatically applies higher guidance scales for complex prompts than for simpler ones and that it can be easily combined with existing guidance schemes such as CFG or LIG. Our code is available at this link.

## 1   Introduction

At the heart of the image and video generation revolution led by diffusion models Sohl-Dickstein et al. [2015], Ho et al. [2020], Song et al. [2021a,b], Karras et al. [2022] lies the conditioning algorithm most well-known as the *guidance* mechanism Dhariwal and Nichol [2021], Ho and Salimans [2021]. The need for diffusion guidance stems from the inherent challenge of learning conditional distributions with limited paired data, which is why further emphasizing the signal using a parameter referred to as the guidance scale proves advantageous in practice Dhariwal and Nichol [2021]. This parameter allows to push the predictions from the conditional model further from those of the unconditional model, essentially reinforcing the difference between them Ho and Salimans [2021], Karras et al. [2024a]. This is more well-known as the Classifier-Free Guidance (CFG) algorithm Ho and Salimans [2021]. It is only after the introduction of the guidance algorithm, that diffusion models gained the impressive performance they are nowadays known for. Recent works have however made clear that applying guidance on the entire sampling trajectory is not only unnecessary but also harms performance Kynkäänniemi et al. [2024], Sadat et al. [2024]. By applying guidance in the beginning of the generative process, the model tends to converge to specific condition-dependent low frequency details Dieleman [2024], Ho and Salimans [2021], Kynkäänniemi et al. [2024], severely reducing the diversity of the generated samples. On the other hand, applying guidance towards the end of the generative process is largely unnecessary as both unconditional and conditional estimates converge,

---

[†] Joint Senior Authors
[*] Corresponding author: *felix.koulischer@ugent.be*

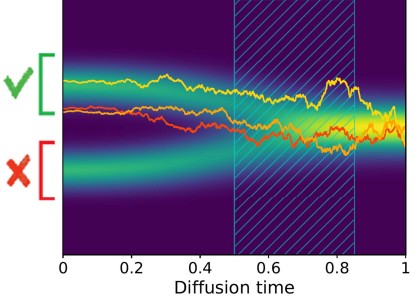 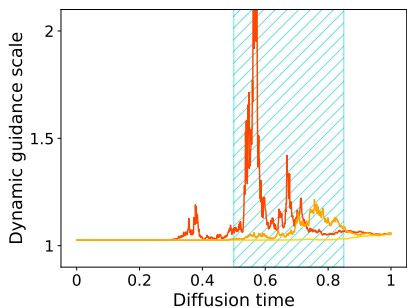

(a) Example conditional diffusion trajectories     (b) Corresponding dynamic guidance scale

Figure 1: Illustrative diffusion trajectories and their hypothetical guidance scales in a 1D setting. Trajectories farther from the mode near the decision window (red, orange) receive stronger guidance, whereas those clearly heading toward the right mode (yellow) receive negligible guidance.

both being equally capable of denoising the high-frequency features Kynkäänniemi et al. [2024], Karras et al. [2024a].

These results can be understood theoretically based on the recently developed theory of spontaneous symmetry breaking phase transitions Raya and Ambrogioni [2023], Sclocchi et al. [2024], Ambrogioni [2025]. This theory predicts that so called 'speciation' transitions during generative diffusion correspond to particular 'decision points' when some features of the generations, such as for example class identity, are maximally sensitive to guidance Biroli et al. [2024], Li and Chen [2024], Handke et al. [2025]. Since the timing of these critical windows depends on both the data and the conditioning signal, it is natural to consider dynamic forms of guidance where the guidance scale is determined state-dependently.

In this work, we provide a principled methodology to obtain dynamic guidance formulas. Our scheme relies on feedback of a quality estimation of its current predictions, which is why we refer to our scheme as **F**eed**B**ack **G**uidance (FBG) in analogy with control theory. As shown in Fig. 1, if a trajectory is estimated as more likely to be originating from the unconditional model than from the conditional model, the guidance scale increases to correct the error and realign the sample towards the correct class. Similarly to the empirically justified work on Limited Interval Guidance (LIG)Kynkäänniemi et al. [2024], our self-regulated guidance scale is only present at intermediate noise levels, which now follows from first principles.

## 2 Preliminaries

To introduce our FeedBack Guidance scheme (FBG), the working principles behind Denoising Diffusion Probabilistic Models (DDPMs) and Classifier-Free-Guidance (CFG) are required. These are summarized in sections 2.1 and 2.2. In section 2.3, an overview of the current stand of literature with respect to diffusion guidance is given.

### 2.1 Denoising Diffusion Probabilistic Models (DDPM)

Diffusion models Sohl-Dickstein et al. [2015], Ho et al. [2020], Song et al. [2021a,b], Karras et al. [2022] are progressive denoisers, that aim to invert a forward noising schedule by learning the score function, defined as the gradient of the log-likelihood of the marginals $\nabla_{\boldsymbol{x}} \log p_t(\boldsymbol{x}; \theta)$. In the case of Variance Exploding (VE) DDPM, this forward noising process iteratively adds Gaussian noise to a clean data distribution. This way, the underlying data distribution $q(\boldsymbol{x}, 0)$ is progressively noised until a Gaussian of very large variance is obtained, i.e. until after $T$ steps $q(\boldsymbol{x}, T) \sim \mathcal{N}(\boldsymbol{x}; \boldsymbol{0}, \sigma_T^2 \boldsymbol{I})$, with $\sigma_T \gg 1$. For any forward noising schedule $\{\sigma_t\}_{t=1}^T$, whereby $\boldsymbol{x}_{t+1} = \boldsymbol{x}_t + \sigma_{t+1|t}\boldsymbol{\epsilon}_t$ (with $\boldsymbol{\epsilon}_t \sim \mathcal{N}(\boldsymbol{\epsilon}_t; \boldsymbol{0}, \boldsymbol{I})$ and $\sigma_{t+1|t}^2 = \sigma_{t+1}^2 - \sigma_t^2$), the forward process can be directly sampled at any step $t$, according to $q(\boldsymbol{x}_t|\boldsymbol{x}_0) \sim \mathcal{N}(\boldsymbol{x}_t; \boldsymbol{x}_0, \sigma_{t|0}^2 \boldsymbol{I})$, with $\sigma_{t|0}^2 = \sum_{i=0}^{t-1} \sigma_{i+1|i}^2 = \sigma_t^2$. The goal is to learn a score based model, that is able to iteratively reverse this forward process. This backward process can

be decomposed into a Markov chain

$$p(\boldsymbol{x}_{t:T};\theta) = p(\boldsymbol{x}_T)\prod_{i=t+1}^{T} p_i(\boldsymbol{x}_{i-1}|\boldsymbol{x}_i;\theta) \tag{1}$$

Crucially, each step of the Markov chain is by construction approximately Gaussian $p_t(\boldsymbol{x}_{t-1}|\boldsymbol{x}_t;\theta) \sim \mathcal{N}(\boldsymbol{x}_{t-1};\boldsymbol{\mu}_{t-1}(\boldsymbol{x}_t),\sigma^2_{t-1|t}\boldsymbol{I})$ with $\sigma^2_{t-1|t} = \sigma^2_{t-1}(1 - \frac{\sigma^2_{t-1}}{\sigma^2_t})$. Instead of modeling the mean, $\boldsymbol{\mu}_{t-1}$, it is common to train a denoiser that predicts the final denoised output at any diffusion stage $\hat{\boldsymbol{x}}_{0|t} = \mathbb{E}_{q(x_0|x_t)}(x_0)$[1]. In the case of a VE-scheduler, the two are connected by the identity:

$$\boldsymbol{\mu}_{\boldsymbol{\theta},t-1} = \frac{\sigma^2_{t-1}}{\sigma^2_t}\boldsymbol{x}_t - \left(1 - \frac{\sigma^2_{t-1}}{\sigma^2_t}\right)\hat{\boldsymbol{x}}_{0|t} \tag{2}$$

It should also be noted that the sought after score funtion is proportional to the learned denoiser.

## 2.2 Classifier-Free Guidance

Learning a highly complex conditional distribution, such as that required for Text-To-Image (T2I), is a challenging task. In most situations it is only possible to poorly approximate this target distribution, which can be seen from the vague predictions generatedby sampling a "pure" conditionally trained diffusion model, such as the one present in Stable diffusion Rombach et al. [2022]. To amplify the conditioning signal present during the denoising process, diverse solutions exist. The most widely used method is that of diffusion guidance, of which the simplest example is that of Classifier (-Free) Guidance Dhariwal and Nichol [2021], Ho and Salimans [2021]. The guidance mechanism is used in all variants of diffusion, from Flow Matching Lipman et al. [2023], Zheng et al. [2023] to Discrete Diffusion Models Schiff et al. [2025]. The reasoning behind guidance is to sharpen the marginals towards the the posterior likelihood $p_{\theta,t}(c|\boldsymbol{x}_t)$ using an exponent $\lambda$ referred to as the guidance scale, i.e. to consider a $\lambda$-sharpened conditional marginal distribution $\tilde{p}_t(\boldsymbol{x}_t|c) = p_{\theta,t}(\boldsymbol{x}_t)p_{\theta,t}(c|\boldsymbol{x}_t)^\lambda$. When $\lambda$ is equal to one this reduces to sampling the conditional model, while setting $\lambda > 1$ further sharpens the marginals towards regions that better satisfy the condition $c$. To obtain a Classifier-Free scheme, independent of the posterior $p_{\theta,t}(c|\boldsymbol{x}_t)$, it suffices to rewrite the posterior probability as a ratio of the conditional and unconditional likelihoods $p_{\theta,t}(c|\boldsymbol{x}_t) \propto p_{\theta,t}(\boldsymbol{x}_t|c)/p_{\theta,t}(\boldsymbol{x}_t)$. From a score-based perspective this results in the well known guidance equation:

$$\begin{aligned}\nabla_{\boldsymbol{x}}\log\tilde{p}_t(\boldsymbol{x}_t|c) &= \nabla_{\boldsymbol{x}}\log p_{\theta,t}(\boldsymbol{x}_t) + \lambda\nabla_{\boldsymbol{x}}\log p_{\theta,t}(c|\boldsymbol{x}_t) \\ &= \nabla_{\boldsymbol{x}}\log p_{\theta,t}(\boldsymbol{x}_t) + \lambda\big(\nabla_{\boldsymbol{x}}\log p_{\theta,t}(\boldsymbol{x}_t|c) - \nabla_{\boldsymbol{x}}\log p_{\theta,t}(\boldsymbol{x}_t)\big)\end{aligned} \tag{3}$$

In the first formulation the gradient of a classifier appears, if this likelihood is directly parametrised using pretrained networks one obtains what is refered to as training-free guidance Dhariwal and Nichol [2021], Shen et al. [2024a]. The main challenge of these approaches is that they leverage a model trained solely on clean samples to offer guidance on noisy samples Shen et al. [2024a], Ye et al. [2024]. The second equation is that of Classifier-Free Guidance Ho and Salimans [2021], which has as main inconvenient that it requires joint conditional-unconditional model training Ho and Salimans [2021], Rombach et al. [2022], Karras et al. [2022, 2024b].

## 2.3 Related Work

Conditional generation, and in particular the topic of guidance, is a very active research field. As this paper is centered around Classifier-Free Guidance and its derivatives in the context of Diffusion Models, this section will provide a concise exploration of the field's key developments, with readers seeking comprehensive context directed to existing in-depth survey literature Anonymous [2024], Adaloglou and Kaiser [2024]. A prominent research direction preserves the core framework of CFG while introducing various predefined time-varying profiles to replace the rigid constant guidance scale Sadat et al. [2024], Kynkäänniemi et al. [2024], Wang et al. [2024], Xia et al. [2024]. Noteworthy, due to its computational advantage, simplicity and its effectiveness, is choosing a limited interval in which guidance should be applied Kynkäänniemi et al. [2024]. Precisely where this limited interval is located depends on the underlying quality of the conditional model. The better the model, as is

---

[1]Equivalently, the noise $\hat{\boldsymbol{\epsilon}}_t$ can be predicted, the two are connected by the identity $\boldsymbol{x}_t = \hat{\boldsymbol{x}}_{0|t} + \sigma_t\hat{\boldsymbol{\epsilon}}_t$

the case for the EDM2 models on which that particular paper focuses, the later the guidance can be activated. Different intervals are found when evaluating performance using the FID Heusel et al. [2017] or FD$_{\text{DinoV2}}$ Stein et al. [2023], the latter being much wider and earlier.

Less researched is the approach of a state-dependent guidance scale, which has been shown to be theoretically optimal in the context of negative guidance Koulischer et al. [2025], Kim et al. [2025], but has only been heuristically proposed for positive guidance Brack et al. [2023], Shen et al. [2024b]. Our work is the first to derive a state- and time-dependent guidance scale from first principles.

Another advance is autoguidance, which replaces the expensive unconditional model by a much weaker one Karras et al. [2024a]. The weaker model may be a smaller version of the same architecture, an undertrained model, or one incorrectly conditioned at earlier timesteps Karras et al. [2024a], Kaiser et al. [2024], Li et al. [2024]. This approach reinforces not only class-specific details but also image quality, relying on the smaller model's errors being of similar nature as those of the stronger one, simply larger. Our Feedback scheme can be adapted to produce equations fully compatible with these principles.

A newly emerging research direction, has proposed to step away from guidance as a whole and instead perform a tree-search over a limited amount of trajectories and selectively choose the most promising ones using reward models Guo et al. [2025], Ma et al. [2025]. These approaches are still very recent, but might lead to an entire different era of conditional diffusion generation, in which instead of relying on guidance, the model has the ability to focus on the most promising trajectories to obtain the most of the learned models. Our feedback approach, and in particular the posterior estimation algorithm, could potentially provide a meaningful way of ranking these paths using solely the pretrained diffusion models.

## 3 Feedback Guidance

This section presents the theoretical foundation of Feedback Guidance. In section 3.1, we introduce a framework that reformulates guidance formulas as the result of assumptions about systematic errors in the learned distributions. In section 3.2 we demonstrate that adopting an additive rather than multiplicative error assumption naturally produces a dynamic guidance mechanism. The resulting state- and time-dependent guidance scale requires posterior likelihood estimation, outlined in section 3.3. In section 3.4 the key hyperparameters of FBG are discussed.

### 3.1 Interpreting guidance schemes through error assumptions

Here, we conceptualize guidance formulas as the result of inverting an error model that determines how the unconditional distribution 'corrupts' the estimated conditional distribution. This form of corruption is to be expected since the model is far less frequently trained on a given class or prompt, which implies that a large part of the training signal is unconditional. In the case of CFG, by reversing Eq. (3) and substituting $\gamma = 1/\lambda$ it becomes clear that CFG implicitly assumes that the learned conditional distribution, denoted by the subscript $\theta$ in this work $p_{\theta,t}(\boldsymbol{x}_t|c)$, is a multiplicative mixture of the true conditional and unconditional distributions:

$$
\begin{aligned}
p_{\theta,t}(\boldsymbol{x}_t|c) &\propto p_t(\boldsymbol{x}_t)^{\frac{\lambda-1}{\lambda}} p_t(\boldsymbol{x}_t|c)^{\frac{1}{\lambda}} \\
&= p_t(\boldsymbol{x}_t)^{1-\gamma} p_t(\boldsymbol{x}_t|c)^{\gamma}
\end{aligned}
\tag{4}
$$

In the case when $\gamma \approx 1$ (low guidance regime), the modelled conditional corresponds to the true conditional. In that setting, sampling the learned conditional is sufficient. However, when $\gamma \approx 0$

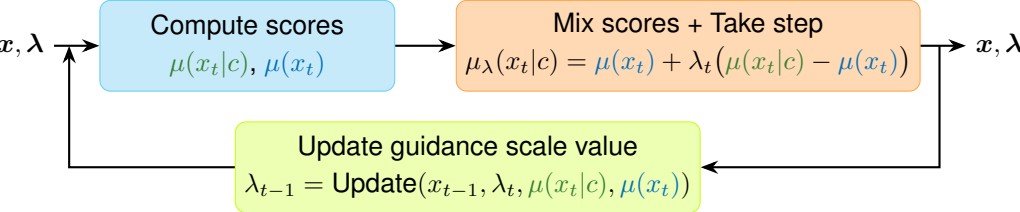

Figure 2: Schematic of Feedback guidance (FBG). The state space consists of both $\boldsymbol{x}_t$ and $\lambda$, which are updated iteratively during the denoising process. The guidance scale is updated by tracking the posterior ratio thanks to Eq. (11), which can then be inserted in Eq. (8).

(strong guidance regime), the modelled conditional resembles the *unconditional* distribution, which is precisely why strong guidance is required during sampling.

## 3.2 Feedback Guidance

In control theory jargon, both standard CFG and LIG are examples of open-loop controllers since the guidance scale is not a function of the current state $\boldsymbol{x}_t$. This means that the guidance formula will equally affect all states, regardless of their quality and class alignment. We argue that might lead to over-saturation and stereotypical generations in situations where the conditional model is already good enough to be sampled on its own, as is the case for simplistic or memorized prompts.

Here, we derive a guidance formula that implements a form of feedback, or closed-loop control. From an error perspective we assume that the learned conditional distribution corresponds to an additive mixture of the true conditional and unconditional distributions:

$$p_{\theta,t}(\boldsymbol{x}_t|c) = (1-\pi)p_t(\boldsymbol{x}_t) + \pi p_t(\boldsymbol{x}_t|c) \tag{5}$$

The additive assumption can be seen as less restrictive than the multiplicative one as it allows the learned conditional distribution $p_{\theta,t}(\boldsymbol{x}_t|c)$ to be non-zero in regions where the true conditional distribution $p_t(\boldsymbol{x}_t|c)$ is zero, a feat the multiplicative assumption is incapable of. Due to the joint training pipeline, and the fact that training pairs often contain more than a single element, such an overlap of the learned distributions is in practice highly likely. Assuming a well-modeled unconditional distribution, i.e. $p_{\theta,t}(\boldsymbol{x}_t) \approx p_t(\boldsymbol{x}_t)$, this implies sampling:

$$p_t(\boldsymbol{x}_t|c) \propto p_{\theta,t}(x|c) - (1-\pi)p_{\theta,t}(x) \tag{6}$$

In other words, we propose removing a portion of the unconditional distribution from the modeled conditional before sampling. This, similarly to the approach taken in CFG, helps strengthen the conditioning signal, as regions that do not satisfy $c$ are pushed towards zero-likelihood.

Of key interest for sampling is the score function of the underlying conditional distribution $\nabla_x \log p_t(x|c)$, which can be derived using the chain rule[2]:

$$\nabla_{\boldsymbol{x}} \log p(\boldsymbol{x}_t|c) = \nabla_{\boldsymbol{x}} \log p_{\theta,t}(\boldsymbol{x}_t) + \lambda(\boldsymbol{x}_t, t)\Big(\nabla_{\boldsymbol{x}} \log p_{\theta,t}(\boldsymbol{x}_t|c) - \nabla_{\boldsymbol{x}_t} \log p_{\theta,t}(\boldsymbol{x}_t)\Big), \tag{7}$$

with as guidance scale:

$$\lambda(\boldsymbol{x}_t, t) = \frac{p_{\theta,t}(c|\boldsymbol{x}_t)/p_{\theta,t}(c)}{p_{\theta,t}(c|\boldsymbol{x}_t)/p_{\theta,t}(c) - (1-\pi)} . \tag{8}$$

The additive error model results in a state- and time-dependent guidance scale that can be expressed in terms of the posterior likelihood $p_{\theta,t}(c|\boldsymbol{x}_t)$. The guidance scale is equal to one when the posterior likelihood is high, and exhibits an asymptotic behavior as the posterior approaches $1-\pi$. The mixing parameter $\pi$ determines when guidance is deemed necessary: if $\pi$ is close to one, indicating a well learned distribution, guidance is only activated when $p_{\theta,t}(c|\boldsymbol{x}_t)$ reaches very low values. In contrast, for poorly learned distributions, with smaller values of $\pi$, guidance is easily activated as soon as the posterior decreases. It should be noted that if $0 < p_{\theta,t}(c|\boldsymbol{x}_t) < 1-\pi$ the guidance scale is negative. We argue that in the continuous case this situation would never arise since the guidance scale would need to cross the asymptote, which should in turn increase the posterior. To avoid this happening in the discretized case, we clamp the posterior to a minimum value $p_{\min}$, which in practice implies clamping the guidance scale at $\lambda_{\max}$[3].

In practice, our dynamic guidance scale defined by Eq. (8) can easily be added on top of any pre-existing guidance method such as CFG or LIG. To make it clear which methods are used, we introduce the following notation: $\text{FBG}_{\text{pure}}$ corresponds to using solely Eq. (8) as guidance scale, $\text{FBG}_{\text{CFG}}$ and $\text{FBG}_{\text{LIG}}$ respectively correspond to adding some base CFG or LIG on top. We refer to FBG as all approaches that use variants of Eq. (8) for guidance. The corresponding error models assumed for these schemes are given in Appendix F.3.

---

[2]A detailed derivation is provided in Appendix A

[3]The two are connected by the identity $p_{\min} = \log\left((1-\pi)\frac{\lambda_{\max}}{\lambda_{\max}-1}\right)$, for more details see Appendix C.2

### 3.3 Posterior approximation by tracking the Markov Chain

Our novel dynamic guidance scale $\lambda(\boldsymbol{x}, t)$ relies on the posterior likelihood $p_{\theta,t}(c|\boldsymbol{x})$, which is in general not available using score-based models. Leveraging recent ideas by from Koulischer et al. [2025], we approximate the required posterior $p(c|\boldsymbol{x}_t)$ by estimating the required likelihoods by tracking the diffusion Markov Chain, defined by Eq. (1) during the denoising process. Key for this estimation is that the likelihood ratio between the conditional and unconditional models can be updated iteratively through:

$$\log p_{\theta,t}(c|\boldsymbol{x}_{t:T}) = \log p_{\theta,t+1}(c|\boldsymbol{x}_{t+1:T}) + \log p_{\theta,t}(\boldsymbol{x}_t|\boldsymbol{x}_{t+1}, c) - \log p_{\theta,t}(\boldsymbol{x}_t|\boldsymbol{x}_{t+1}) \tag{9}$$

Both markov transitions likelihoods are parametrised as gaussians, resulting in:

$$\log p_{\theta,t}(c|\boldsymbol{x}_t) = \log p_{\theta,t+1}(c|\boldsymbol{x}_{t+1}) - \frac{1}{2\sigma_{t|t-1}^2}\left(\|\boldsymbol{x}_t - \boldsymbol{\mu}_{\theta,t}(\boldsymbol{x}_{t+1}|c)\|^2 - \|\boldsymbol{x}_t - \boldsymbol{\mu}_{\theta,t}(\boldsymbol{x}_{t+1})\|^2\right) \tag{10}$$

This equation estimates the posterior by comparing conditional and unconditional model performance at each denoising step, effectively computing a likelihood ratio weighted by the inverse noise variance $\sigma_{t|t-1}^2$. As the transition kernel sharpens, the posterior estimates become increasingly decisive, allowing for more abrupt shifts in the probability assessment. A crucial advantage of computing the posterior using the scheme describe above is that it causes negligible computational overhead as all required quantities, in particular $\boldsymbol{\mu}_{\theta,t}(\boldsymbol{x}_{t+1})$ and $\boldsymbol{\mu}_{\theta,t}(\boldsymbol{x}_{t+1}|c)$, are already computed.

By tracking the posterior likelihood during inference, we estimate a state- and time-dependent guidance scale and feed it back to the denoiser. Crucially, posterior computation and denoising are staggered, effectively solving a joint ODE–SDE system Skreta et al. [2025], Karczewski et al. [2025]. The closed-loop diagram shown in Fig. 2 and described in detail in Alg. 1 summarizes our approach. This control diagram is progressively unrolled during the denoising process, implying a succession of computing the score functions, mixing them, applying a denoising step, updating the value of the guidance scale and repeating a fixed amount of times until a fully denoised image is obtained.

### 3.4 Defining Practical Hyperparameters

The previously described posterior likelihood estimation via Eq. (10) however suffers from a self-reference bias: when using the conditional model's own prediction ($\boldsymbol{x}_t = \boldsymbol{\mu}_{t,\theta}(\boldsymbol{x}_{t+1}|c)$) as the sampling trajectory, the model effectively evaluates its performance on its own output, artificially inflating its perceived accuracy. This creates a circular reasoning problem where the conditional model always appears superior because it evaluates a trajectory it created. We address this by introducing a linear bias term $-\delta$ (Eq. 11), which allows the unconditional model's predictions to receive appropriate consideration. This adjustment forces the system to recognize that the conditional model's apparent superiority stems from self-comparison rather than objective performance, creating a more balanced sampling process that better represents the true posterior distribution. In practice, this forces the posterior to decrease in the early stages of diffusion, enabling the activation of guidance.

$$\log p_t(c|\boldsymbol{x}_t) = \log p_{t+1}(c|\boldsymbol{x}_{t+1}) - \frac{\tau}{2\sigma_t^2}\left(\|\boldsymbol{x}_t - \boldsymbol{\mu}_{t,\theta}(\boldsymbol{x}_{t+1}|c)\|^2 - \|\boldsymbol{x}_t - \boldsymbol{\mu}_{t,\theta}(\boldsymbol{x}_{t+1})\|^2\right) - \delta \tag{11}$$

The complex non-linear interplay between the three hyperparameters of our approach -$\pi$, $\tau$, and $\delta$- makes it challenging to predict how changing one parameter affects the overall guidance profile. To address this issue, we propose a more intuitive parameterization that allows users to directly control the characteristics of the guidance profile through normalized diffusion timesteps. Instead of directly tuning $\tau$ and $\delta$, we express them as functions of two more interpretable parameters: $t_0$ and $t_1$. Here, $t_0$ represents the normalized diffusion time at which the guidance scale reaches a predefined reference value $\lambda_{\text{ref}}$ (set to 3 without loss of generality), while $t_1$ represents an estimant of the normalized diffusion time at which the guidance reaches its maximum value. Details regarding the hyperparameters are provided in Appendix C.2.

| Guidance scheme | FID (↓) | | FD$_{\text{Dinov2}}$ (↓) | | Prec. (↑) | | Rec. (↑) | |
|---|---|---|---|---|---|---|---|---|
| | Stoch. | PFODE | Stoch. | PFODE | Stoch. | PFODE | Stoch. | PFODE |
| CFG | 5.00 | 2.97 | 100.2 | 88.4 | 0.85 | 0.84 | 0.73 | 0.75 |
| Weight scheduler | 4.58 | 2.75 | 103.1 | 97.1 | 0.84 | 0.83 | 0.74 | 0.76 |
| CFG++ | / | 3.66 | / | 87.8 | / | 0.86 | / | 0.73 |
| LIG | **3.59** | **_2.31_** | 88.5 | 77.1 | 0.86 | 0.86 | 0.75 | 0.77 |
| FBG$_{\text{pure}}$ (ours) | 3.76 | 2.50 | 89.0 | 75.6 | 0.86 | **0.86** | **0.76** | **_0.77_** |
| FBG$_{\text{LIG}}$ (ours) | 3.62 | 2.45 | **87.9** | **_74.6_** | **_0.87_** | 0.86 | 0.75 | 0.76 |

Table 1: Evaluation of different guidance methods using EDM2-XS using a stochastic ('Stoch.') and a 2$^{\text{nd}}$-order Heun sampler ('PFODE'). FID and FD$_{\text{Dinov2}}$ values refer to the model optimized under the respective metrics. Precision and Recall are computed on 10,240 samples with 5 nearest neighbour with models optimized under FD$_{\text{Dinov2}}$ Kynkäänniemi et al. [2019], Stein et al. [2023].

## 4 Results

We validate our novel state-dependent dynamic guidance scheme on ImageNet512×512 using EDM2 models, where it consistently outperforms CFG and remains competitive with LIG, while benefitting from a strong mathematical framework. To assess generality, we additionally compute FIDs on MS-COCO in the T2I setting. Alongside these quantitative results, we provide qualitative examples showing that the self-regulated feedback guidance scale naturally increases with prompt complexity.

### 4.1 Comparison of Guidance Schemes on Class Conditional Generation

For class-conditional experiments, we use EDM2-XS Karras et al. [2024b,a] trained on ImageNet512×512 Deng et al. [2009] with 64 function evaluations. Performance is measured via FID Heusel et al. [2017], FD$_{\text{DinoV2}}$ Stein et al. [2023], and Precision/Recall Kynkäänniemi et al. [2019], providing a comprehensive view of the quality–diversity trade-off Kynkäänniemi et al. [2019], Stein et al. [2023], Kynkäänniemi et al. [2024]. Consistent with our posterior estimation framework based on Markov chain sampling, we focus on stochastic samplers but note similar results using the probability flow ODE with a 2$^{\text{nd}}$-order Heun solver Karras et al. [2022, 2024b]. Baselines are optimized per sampler: CFG Ho and Salimans [2021], CFG++ Chung et al. [2025] and the linear guidance weight scheduler Wang et al. [2024] via a grid search over the guidance scale, and LIG Kynkäänniemi et al. [2024] via joint search over $\sigma_{\max}$ and guidance scale, followed by $\sigma_{\min}$ tuning[4]. For Feedback Guidance, we sweep $t_0$, $t_0 - t_1$, and $\pi$. We note that CFG++ was originally only tested on text-to-image generation Chung et al. [2025], while the linear guidance weight scheduler was only benchmarked using the FID as metric Wang et al. [2024].

To visualize parameter effects, we present FD$_{\text{DinoV2}}$ sweeps as a heatmap in Fig. 3 (FID results in Appendix F). Optimal hyperparameters are listed in Table 3 and vary across metrics (FID vs. FD$_{\text{DinoV2}}$) and samplers. Consistent with LIG, optimal FD$_{\text{DinoV2}}$ performance requires earlier and longer guidance activation, corresponding to larger $t_0$ and $t_0 - t_1$ for FBG.
FBG$_{\text{pure}}$ outperforms CFG, CFG++ and the linear scheduler on both FD$_{\text{DinoV2}}$ and FID, while remaining competitive with LIG. To assess the quality–diversity trade-off, we compute Precision–Recall curves on 10,240 images with 5 nearest neighbor with optimal FD$_{\text{DinoV2}}$ settings. CFG and LIG are swept over guidance scales 1–4, while FBG$_{\text{pure}}$ is swept over $t_0$ with fixed $t_0 - t_1 = 0.125$ (8/64 steps). Results confirm that CFG improves quality at a large cost to diversity, LIG better preserves diversity due to its narrow late-stage guidance, and FBG achieves CFG-like Recall but with substantially higher Precision, offering a better quality–diversity balance.
Finally, we optimize FBG$_{\text{LIG}}$ by fixing the LIG interval and solely varying the guidance scale, with $\pi$ and $t_0$ taken from FBG$_{\text{pure}}$ and $t_1$ adjusted. While a full grid search could further improve results, this hybrid outperforms its components on FD$_{\text{DinoV2}}$, which aligns closely with human perception Stein et al. [2023], highlighting the complementarity of our approach. All optimized metrics are provided in Table 1 and additional details on the various methods are provided in Appendix F. Best performing metrics per sampler are written in bold and best performing overall are further underlined.

---

[4]Contrary to Kynkäänniemi et al. [2024], we find that late-stage guidance is not only unnecessary but also harmful, particularly for FD$_{\text{DinoV2}}$.

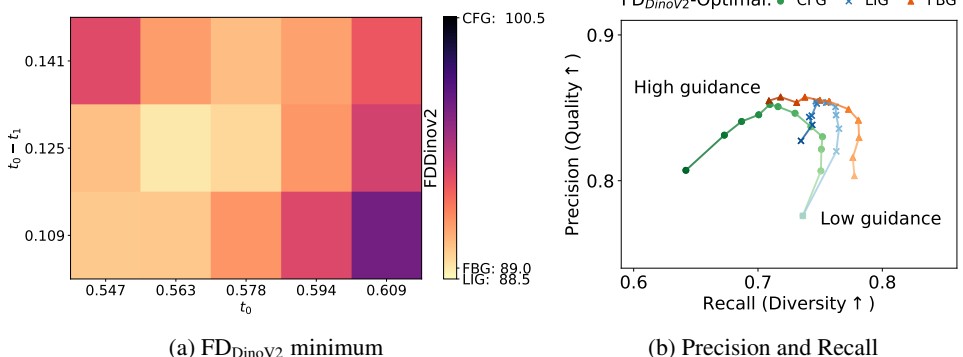

|  | (a) FD$_{\text{DinoV2}}$ minimum | (b) Precision and Recall |
|---|---|---|

Figure 3: (a) Grid search over $t_0$ and $t_1$, with FD$_{\text{DinoV2}}$ calibrated to the best value among CFG, LIG, and FBG. (b) Precision–Recall sweeps at each method's FD$_{\text{DinoV2}}$ optimum: CFG/LIG sweep guidance scale, FBG sweeps $t_0$ at fixed $t_0 - t_1$. Guidance strength is indicated by color intensity.

## 4.2 Guidance on Text-To-Image

In the context of Text-To-Image (T2I), our approach is evaluated using Stable diffusion 2 Rombach et al. [2022], for which a VE-scheduler is implemented Karras et al. [2022, 2024b]. To remain consistent with the theory a stochastic sampler using 32 function evaluations is used. The purpose of this section is not to investigate to what extent Feedback Guidance may outperform CFG or LIG in terms of image quality, but merely to demonstrate the promise of the approach.

On its own, the conditional model used in T2I applications is of far lower quality than is the case for the EDM2 models, which is precisely why in practice much larger guidance scales are required Ho and Salimans [2021], Rombach et al. [2022]. For FBG this implies that $\pi$ has to be chosen much smaller, all images shown in this section use a value of $\pi = 0.85$ in combination with $t_0 = 0.75$ and $t_1 = 0.5$. In practice, we also find it helpful to remove the offset from the posterior approximation towards the end of the generative process[5]. We find that using a limited amount of CFG with $\lambda_{\text{CFG}} = 1.5$ can help to retrieve low frequency features such as sharp colors, without significantly harming diversity, which is why in this context we propose to use FBG$_{\text{CFG}}$ for visual evaluation. To assess our approach beyond visual inspection, we compute FID, FD$_{\text{DinoV2}}$ and Aesthetic-Score[6] on 3k MS-COCO prompts Lin et al. [2014], Heusel et al. [2017], Oquab et al. [2024]. Results are given in Tab. 2[7] and show similar trends to those from the class-conditional setting, underscoring the method's generality. Although MS-COCO captions are relatively simple and thus less ideal for testing guidance methods, most effective on complex prompts, it remains the standard benchmark for qualitative T2I evaluation.

| Guidance Scheme | FID (↓) | FD$_{\text{DinoV2}}$ (↓) | Aesthetic Score (↑) |
|---|---|---|---|
| CFG | 19.64 | 54.56 | 5.65 |
| LinCFG | 19.15 | 53.26 | 5.71 |
| LIG | 18.81 | 54.25 | 5.74 |
| FBG$_{\text{pure}}$ (ours) | **18.63** | 53.11 | 5.75 |
| FBG$_{\text{CFG}}$ (ours) | 18.63* | **52.14** | **5.75** |

Table 2: Comparison of methods across FID, FD$_{\text{DinoV2}}$, and Aesthetic Score. Evaluated using 3k prompts from the MS-COCO dataset Lin et al. [2014] using SDv2 Rombach et al. [2022].

We now emphasize two key, novel, properties of our feedback guidance approach:

**Prompt specificity of FBG:** A key advantage of Feedback Guidance is that it adapts denoising behavior per prompt, unlike fixed-profile methods that treat all prompts identically. Well-learned prompts (e.g., "The Starry Night") should receive minimal guidance, whereas complex, descriptive

---

[5]To be specific in this context we choose $\delta = 0$ if $t < 0.3$.

[6]Model accesible at https://github.com/discus0434/aesthetic-predictor-v2-5

[7]The optimised hyperparameters are given in Table 4

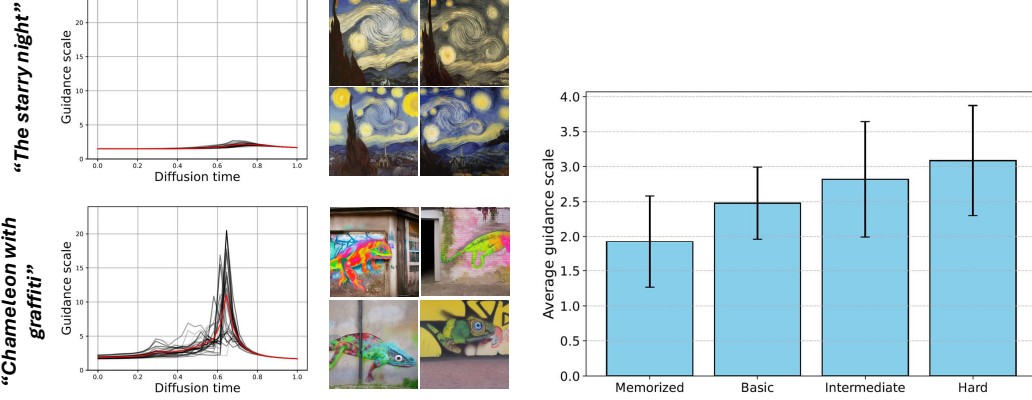

(a) Examples of dynamic guidance scale     (b) Average guidance for various prompt difficulties

Figure 4: Analysis of FBG in the context of T2I. In (a) the dynamic guidance scale of 32 samples are shown using two prompts: a memorized one ( *"The starry night by Van Gogh"*) and a more difficult one ( *"A chameleon blending into a graffiti-covered wall"*). In (b) the average guidance scale applied when using FBG is shown as a function of various prompt difficulties specified in Appendix G.

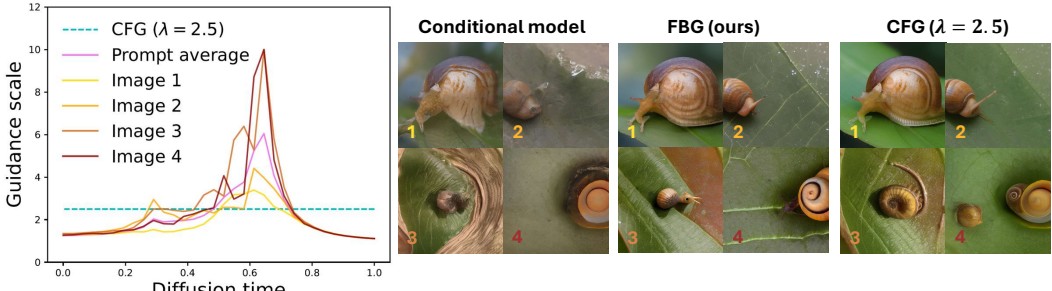

Figure 5: Guidance scale for different trajectories using the prompt: *"A snail crawling on a green leaf with water droplets".* If the conditional prediction is good the guidance is low (top two images). In contrast when the conditional prediction is poor, the guidance scale increases (bottom two images).

prompts require stronger guidance. To test this, we construct a 60-prompt dataset with four difficulty levels: memorized, easy, intermediate, and very hard. More detail on this dataset are provided in Appendix G. We report the average guidance scale obtained with 32 samples per prompt and find, as shown in Fig. 4b, that FBG applies the strongest guidance for the most difficult prompts, as expected. We provide further examples in Appendix F.5.

**Trajectory specificity of FBG:** Another key property of FBG is its state, or trajectory, dependence. For the same prompt, two trajectories might receive entirely different levels of guidance. This is illustrated in Fig. 5: if the conditional model is, by chance, already close to the desired result, a minimal amount of guidance is applied, whereas if the conditional model is far off guidance is much more present.

## 5   Limitations and Future Work

Our FBG approach opens several new directions in both theory and practice of guidance. The adopted additive mixture model was chosen based on its mathematical simplicity, but it is likely not a close approximation of the true systematic biases in trained models. More precise error models could potentially be obtained by studying the training dynamics of the models and tracking the relative error of the learned conditional and unconditional scores, which may lead to more accurate

dynamic guidance formulas. Similarly, our choice of prior is solely based on simplicity, and other options should be investigated both theoretically and empirically. Our evaluation currently focuses on EDM2-XS models. Future work should investigate larger architectures such as EDM2-L or DiT Peebles and Xie [2022], and extend quantitative T2I evaluation to broader prompt datasets, such as LAION5B Schuhmann et al. [2022], to better assess guidance performance across different prompt complexities.

# 6    Conclusion

In this work, we introduced a novel view of the commonly used guidance mechanism, interpreting guidance as a way of rectifying the errors made by the learned conditional model. By replacing the implicitly assumed multiplicative error of Classifier-Free Guidance (CFG) with an additive one, we obtained Feedback Guidance (FBG), a state- and time-dependent guidance mechanism that dynamically relies on the model's own prediction to estimate how much guidance is needed during inference. This work challenges the view of guidance as a fixed global scheme and instead allows different trajectories and conditions to behave differently. Our results demonstrate that FBG significantly outperforms CFG, and is competitive with Limited Interval Guidance (LIG) while relying on solid theoretical grounds.

**Acknowledgments**

This research was partly funded by the Research Foundation - Flanders (FWO-Vlaanderen) under grant G0C2723N and by the Flemish Government (AI Research Program).

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

## A  Detailed derivation of Feedback guidance scheme

Our Feedback Guidance scheme is based on the assumption that the learned conditional distribution $p_{\theta,t}(\boldsymbol{x}_t|c)$ can be expressed as an *additive mixture* of the true conditional and unconditional distributions, $p_t(\boldsymbol{x}_t|c)$ and $p_t(\boldsymbol{x}_t)$, with mixing coefficient $\pi$:

$$p_{\theta,t}(\boldsymbol{x}_t|c) = \pi p_t(\boldsymbol{x}_t|c) + (1-\pi)p_t(\boldsymbol{x}_t). \tag{12}$$

Our objective is to sample from the true conditional distribution. Assuming a well-learned unconditional model, this can be written as:

$$p_t(\boldsymbol{x}_t|c) \propto p_{\theta,t}(\boldsymbol{x}_t|c) - (1-\pi)p_t(\boldsymbol{x}_t). \tag{13}$$

For score-based generative models, the relevant quantity is the score function $\nabla_{\boldsymbol{x}} \log p_t(\boldsymbol{x}_t|c)$. Applying the chain rule and using $\nabla_{\boldsymbol{x}}p(\boldsymbol{x}) = p(\boldsymbol{x})\nabla_{\boldsymbol{x}} \log p(\boldsymbol{x})$, we obtain:

$$\begin{aligned}
\nabla_{\boldsymbol{x}} \log p_t(\boldsymbol{x}_t|c) &= \nabla_{\boldsymbol{x}} \log \left[ p_{\theta,t}(\boldsymbol{x}_t|c) - (1-\pi)p_t(\boldsymbol{x}_t) \right] \\
&= \frac{\nabla_{\boldsymbol{x}}p_{\theta,t}(\boldsymbol{x}_t|c) - (1-\pi)\nabla_{\boldsymbol{x}}p_t(\boldsymbol{x}_t)}{p_{\theta,t}(\boldsymbol{x}_t|c) - (1-\pi)p_t(\boldsymbol{x}_t)} \\
&= \frac{p_{\theta,t}(\boldsymbol{x}_t|c)\nabla_{\boldsymbol{x}} \log p_{\theta,t}(\boldsymbol{x}_t|c) - (1-\pi)p_t(\boldsymbol{x}_t)\nabla_{\boldsymbol{x}} \log p_t(\boldsymbol{x}_t)}{p_{\theta,t}(\boldsymbol{x}_t|c) - (1-\pi)p_t(\boldsymbol{x}_t)} \\
&= \nabla_{\boldsymbol{x}} \log p_t(\boldsymbol{x}_t) + \frac{p_{\theta,t}(\boldsymbol{x}_t|c)}{p_{\theta,t}(\boldsymbol{x}_t|c) - (1-\pi)p_t(\boldsymbol{x}_t)}\left(\nabla_{\boldsymbol{x}} \log p_{\theta,t}(\boldsymbol{x}_t|c) - \nabla_{\boldsymbol{x}} \log p_t(\boldsymbol{x}_t)\right).
\end{aligned} \tag{14}$$

Defining the feedback guidance scale $\lambda(\boldsymbol{x}_t, t)$ through Eq. (8), a familiar-looking guidance equation is obtained:

$$\nabla_{\boldsymbol{x}} \log p_t(\boldsymbol{x}_t|c) = \nabla_{\boldsymbol{x}} \log p_t(\boldsymbol{x}_t) + \lambda(\boldsymbol{x}_t, t)\left(\nabla_{\boldsymbol{x}} \log p_{\theta,t}(\boldsymbol{x}_t|c) - \nabla_{\boldsymbol{x}} \log p_t(\boldsymbol{x}_t)\right). \tag{15}$$

This final expression summarizes our approach of feedback guidance. This equation generalizes the typical classifier-free guidance scheme to possess a both state- and time-dependent guidance scale $\lambda(\boldsymbol{x}_t, t)$, enabling adaptive control over the denoising process.

## B  Theoretical shortcomings of CFG resolved by FBG

Despite being standardly used in practice, CFG remains vastly misunderstood Bradley and Nakkiran [2024], Chidambaram et al. [2024]. The common loose understanding of guidance is that to reinforce the conditioning signal of the models the aim is to sample from a $\lambda$-sharpened ditribution, i.e. from $\tilde{p}(x|c) = p(x)p(c|x)^\lambda$. From this it is typical to derive $\nabla_{\boldsymbol{x}} \log \tilde{p}(x|c)$ and obtain the standard linear combination of conditional and unconditional scores used in all variants of guidance Dhariwal and Nichol [2021], Ho and Salimans [2021].

What it is often overlooked is that this mixing of the distribution is defined *locally*, at a specific noise level defied by $t$. By assuming an equivalent multiplicative mixing at every noise level, as for instance the case when using a constant guidance scale in CFG, the sampled marginals do not correspond to the predefined forward process. In essence the common misunderstanding is that the mixing operation commutes with the noising operation, i.e. that mixing the noise-free distributions and noising them is equivalent to first noising the distributions and then only mixing them, which is erroneous. In other words, in the case of a multiplicative mixture one can not simply mix the clean distributions, add a subscript $t$ everywhere and expect these to sample from the desired marginals. When following the score functions defined using the CFG equation, one is not sampling from the intuitively sharpened data distribution $\tilde{p}(x|c) = p(x)p(c|x)^\lambda$ Bradley and Nakkiran [2024], Chidambaram et al. [2024].

To sample the $\gamma$-sharpened conditional distribution in the clean image space using the forward process defined by the diffusion kernel $k(\boldsymbol{x}_s, \boldsymbol{x}_t)$ the marginals at timestep $t$ should correspond to:

$$\begin{aligned}
\tilde{p}_t(\boldsymbol{x}_t|c) &= \int \mathrm{d}\boldsymbol{x}_0 \tilde{p}_0(\boldsymbol{x}_0|c)k(\boldsymbol{x}_0, \boldsymbol{x}_t) \\
&= \int \mathrm{d}\boldsymbol{x}_0 p_{\theta,0}(\boldsymbol{x}_0|c)^\lambda p_{\theta,0}(\boldsymbol{x}_0)^{1-\lambda}k(\boldsymbol{x}_0, \boldsymbol{x}_t)
\end{aligned} \tag{16}$$

However, when sampling using the score function defined by CFG we are implicitly assuming that the marginals at timestep $t$ keep the mixing property:

$$
\begin{aligned}
\tilde{p}_{t,CFG}(\boldsymbol{x}_t|c) &= p_{\theta,t}(\boldsymbol{x}_t|c)^\lambda p_{\theta,t}(\boldsymbol{x}_t)^{1-\lambda} \\
&= \Big( \int \mathrm{d}\boldsymbol{x}_0 \tilde{p}_0(\boldsymbol{x}_0|c) k(\boldsymbol{x}_0, \boldsymbol{x}_t) \Big)^\lambda \Big( \int \mathrm{d}\boldsymbol{x}_0 \tilde{p}_0(\boldsymbol{x}_0) k(\boldsymbol{x}_0, \boldsymbol{x}_t) \Big)^{1-\lambda}
\end{aligned}
\tag{17}
$$

These two expressions are not equal and can in fact differ significantly especially at higher noise levels, at which the overlap between $p_t(x|c)$ and $p_t(x)$ is significant as the information of the underlying distributions has been nearly entirely removed, i.e. both distributions start to ressemble gaussian distributions. This implies that the trajectories sampled under CFG simply do not correspond to the predefined forward process, especially at high noise levels. The main consequence of this is that sampling using CFG can be very misleading.

Mathematically the non-commutability of the mixing and noising operations in the case of CFG is due to the non-commutability of the multiplication and convolution operations. The proposed *additive* error at the heart of our FBG approach does not suffer from the same flaws. Thanks to the commutability of the addition and the convolution, the mixing and noising operations become interchangeable. This implies that when using the scores provided by Feedback guidance the predefined distribution is retrieved[8].

Resolving this issue is not the main purpose of this work, which is why we for instance propose combining FBG with other guidance scheme such as CFGHo and Salimans [2021] or LIGKynkäänniemi et al. [2024]. Nonetheless, we believe this to be an insightful discussion worth mentioning and exploring.

## C   Posterior likelihood estimation

The estimation of the posterior likelihood is central to the proposed Feedback Guidance. Multiple approaches in the literature provide ways of estimating such densities Koulischer et al. [2025], Skreta et al. [2025], Chewi et al. [2025], Karczewski et al. [2025]. Seeing the similarities of the present work with the Dynamic Negative Guidance (DNG) approach, we keep the same notation Koulischer et al. [2025]. For an intuitive explanation of the estimation, we refer the reader to the afore mentioned paper. The continuous limit is then derived in C.1. The key hyperparameters introduced in the main body are described in more detail in C.2.

### C.1   Continuous limit

In this section, we derive the continuous-time limit of the posterior approximation used in Feedback Guidance (FBG). To this end, it suffices to consider the posterior update under the guided prediction: $\boldsymbol{x}_{t-1} = \boldsymbol{\mu}(\boldsymbol{x}_t) + \lambda(\boldsymbol{\mu}(\boldsymbol{x}_t|c) - \boldsymbol{\mu}(\boldsymbol{x}_t)) + \sigma_{t-1|t}\epsilon$ with $\epsilon \sim \mathcal{N}(0, \boldsymbol{I})$. This results in:

$$
\begin{aligned}
\log p(c|\boldsymbol{x}_{t-1}) &= \log p_t(c|\boldsymbol{x}_t) - \frac{1}{2\sigma_{t-1|t}^2}\big( \|\boldsymbol{x}_{t-1} - \boldsymbol{\mu}(\boldsymbol{x}_t|c)\|^2 - \|\boldsymbol{x}_{t-1} - \boldsymbol{\mu}(\boldsymbol{x}_t)\|^2 \big) \\
&= \log p_t(c|\boldsymbol{x}_t) - \frac{1}{2\sigma_{t-1|t}^2}\big( \|\boldsymbol{\mu}(\boldsymbol{x}_t) + \lambda(\boldsymbol{x}_t,t)(\boldsymbol{\mu}(\boldsymbol{x}_t|c) - \boldsymbol{\mu}(\boldsymbol{x}_t)) + \sigma_{t-1|t}\epsilon - \boldsymbol{\mu}(\boldsymbol{x}_t|c)\|^2 \\
&\quad - \|\boldsymbol{\mu}(\boldsymbol{x}_t) + \lambda(\boldsymbol{x}_t,t)(\boldsymbol{\mu}(\boldsymbol{x}_t|c) - \boldsymbol{\mu}(\boldsymbol{x}_t)) + \sigma_{t-1|t}\epsilon - \boldsymbol{\mu}(\boldsymbol{x}_t)\|^2 \big) \\
&= \log p_t(c|\boldsymbol{x}_t) - \frac{\tau}{2\sigma_{t-1|t}^2}\big( \|(\lambda(\boldsymbol{x}_t,t) - 1)(\boldsymbol{\mu}(\boldsymbol{x}_t|c) - \boldsymbol{\mu}(\boldsymbol{x}_t)) + \sigma_{t-1|t}\epsilon\|^2 \\
&\quad - \|\lambda(\boldsymbol{x}_t,t)(\boldsymbol{\mu}(\boldsymbol{x}_t|c) - \boldsymbol{\mu}(\boldsymbol{x}_t)) + \sigma_{t-1|t}\epsilon\|^2 \big) \\
&= \log p_t(c|\boldsymbol{x}_t) - \frac{\tau}{2\sigma_{t-1|t}^2}\big( (1 - 2\lambda(\boldsymbol{x}_t,t))\|\boldsymbol{\mu}(\boldsymbol{x}_t|c) - \boldsymbol{\mu}(\boldsymbol{x}_t)\|^2 + 2\sigma_{t-1|t}\epsilon \cdot (\boldsymbol{\mu}(\boldsymbol{x}_t|c) - \boldsymbol{\mu}(\boldsymbol{x}_t)) \big)
\end{aligned}
\tag{18}
$$

The term added to the log posterior likelihood can be decomposed into two separate contributions. The first corresponds to a deterministic measure, evaluating how much the conditional and unconditional

---

[8]At least in the case that an exact posterior is available and that the to-be-sampled distribution is valid, i.e. satisfies positivity constraints.

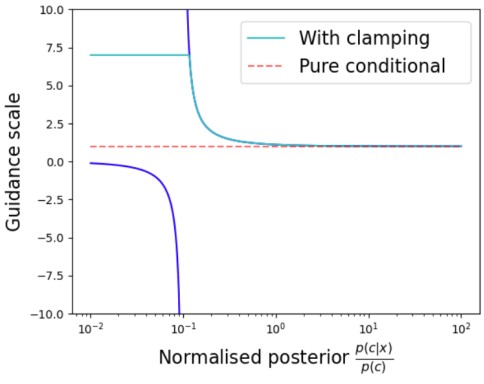

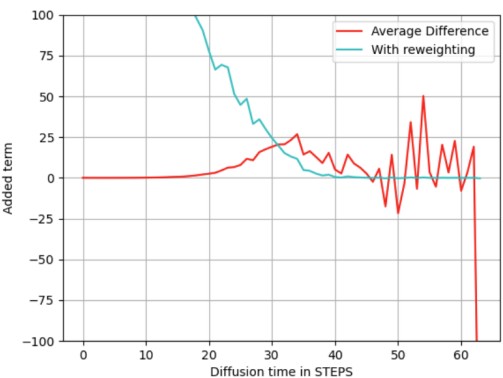

(a) Guidance as function of posterior          (b) Average term added at each timestep

Figure 6: In (a) the guidane scale is plotted as a function of the posterior. Here it is important to note that we argue that in practice the negative part of the curve is never reached as in a continuous process the crossing of the assymptote would result in an infinite amount of guidance. In practice to avoid this happening upon discretisation a maximal guidance value can be set, as shown in the light blue line. In (b) an average of the added term is shown. The red line corresponds to the euclidean difference, whereas the light-blue line includes the reweighting by the transition kernel variance $\sigma_{t-1|t}^2$

predictions differ $\|\boldsymbol{\mu}(\boldsymbol{x}_t) - \boldsymbol{\mu}(\boldsymbol{x}_t)\|^2$, while the second is a stochastic term that measures the overlap between this difference and the added stochastic noise. The former measures how much on average the predictions differ, while the latter measures if by chance the stochastic process has favored one over the other.

Using this understanding, a continuous form of the equations is recognisable. For this one has to realise that in the continuous case the backward and forward transition kernels become of equal variance, i.e. $\lim_{dt \to 0} \sigma_{t-dt|t}^2 = \sigma_{t|t-dt}^2$. Therefore up to first order one can approximate that $\sigma_{t-dt|t}^2 \approx \sigma_{t|t-dt}^2 = \sigma_t^2 - \sigma_{t-1}^2 = d\sigma_t^2/dt = 2\sigma_t \dot{\sigma}_t$. This results in the following continuous integral:

$$\log p(c|\boldsymbol{x}_t) = \int_t^T \frac{\tau}{4\sigma_s \dot{\sigma}_s}(2\lambda(\boldsymbol{x}_s, s) - 1)\|\boldsymbol{\mu}(\boldsymbol{x}_s|c) - \boldsymbol{\mu}(\boldsymbol{x}_s)\|^2 ds + 2\int_t^T \sqrt{2\sigma_s \dot{\sigma}_s}\big(\boldsymbol{\mu}(\boldsymbol{x}_s|c) - \boldsymbol{\mu}(\boldsymbol{x}_s)\big) \cdot d\boldsymbol{W}$$

(19)

The first integral is a path integral, capturing the cumulative deterministic contribution over the diffusion path. The second integral is a stochastic Itô integral over the Wiener process $d\boldsymbol{W}$, capturing how the stochastic noises influence over the posterior likelihood.

### C.2 Understanding the hyperparameters for the posterior estimation

The hyperparameters $\pi$, $\tau$, and $\delta$ play a central role in Feedback Guidance (FBG), but their inter-dependence and abstract nature can hinder accessibility and ease of tuning. To mitigate this, we propose a reparametrization of the temperature and offset parameters, $\tau$ and $\delta$, in terms of the mixing parameter $\pi$ and two new hyperparameters—$t_0$ and $t_1$—which correspond to normalized diffusion times ($t = 0$ denotes clean data; $t = 1$ denotes fully noised data).

Understanding the influence of these parameters requires analyzing how the posterior, and thereby the guidance scale $\lambda(\boldsymbol{x}, t)$, depends on them.

**Effect of mixing parameter $\pi$**

The mixing parameter $\pi$ controls the point at which the guidance scale $\lambda(\boldsymbol{x}, t)$ becomes large. Specifically, $\lambda$ diverges when $p(c|\boldsymbol{x}_t) \approx 1 - \pi$. Thus, adjusting $\pi$ shifts the asymptotic region of the guidance curve. In Fig. 7a, we illustrate this behavior by plotting the posterior values at which $\lambda = 3$ for various values of $\pi$. As $\pi$ increases, stronger confidence (i.e., lower posterior values) is required to activate a given level of guidance. Intuitively, when $\pi \to 1$, the conditional model closely approximates the true posterior, and additional guidance is only necessary when uncertainty is high.

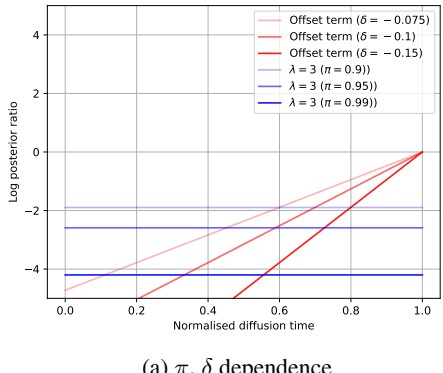 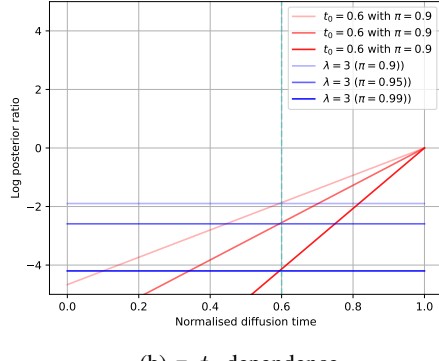



(a) $\pi$, $\delta$ dependence      (b) $\pi$, $t_0$ dependence



Figure 7: Illustration of the interplay between $\pi$ and $\delta$ in (a) and $t_0$ in (b). When specifying $t_0$ the value of $\delta$ is chosen such that the intersection of the red and blue lines is located at $t_0$.

**Motivation for Offset parameter $\delta$**

In the early diffusion steps, when noise dominates, the trajectory of the generated sample is best approximated by the conditional model itself, i.e., $\boldsymbol{x}_{t-1} \approx \boldsymbol{\mu}_{t,\theta}(\boldsymbol{x}t|c)$. Neglecting stochasticity, the posterior update simplifies to:

$$
\begin{aligned}
\log p_t(c|\boldsymbol{x}_t) &= \log p_{t+1}(c|\boldsymbol{x}_{t+1}) - \frac{1}{2\sigma_{t|t+1}^2} \left( \|\boldsymbol{x}_t - \boldsymbol{\mu}_{t,\theta}(\boldsymbol{x}_{t+1}|c)\|^2 - \|\boldsymbol{x}_t - \boldsymbol{\mu}_{t,\theta}(\boldsymbol{x}_{t+1})\|^2 \right) \\
&\approx \log p_{t+1}(c|\boldsymbol{x}_{t+1}) + \frac{1}{2\sigma_{t|t+1}^2} \|\boldsymbol{x}_t - \boldsymbol{\mu}_{t,\theta}(\boldsymbol{x}_{t+1})\|^2
\end{aligned}
\tag{20}
$$

This leads to a monotonically increasing posterior, which artificially inflates model confidence and suppresses guidance activation. To address this, we introduce a linear transformation with an offset $\delta$:

$$
\log p_t(c|\boldsymbol{x}_t) = \log p_{t+1}(c|\boldsymbol{x}_{t+1}) - \frac{\tau}{2\sigma_{t|t+1}^2} \left( \|\boldsymbol{x}_t - \boldsymbol{\mu}_{t,\theta}(\boldsymbol{x}_{t+1}|c)\|^2 - \|\boldsymbol{x}_t - \boldsymbol{\mu}_{t,\theta}(\boldsymbol{x}_{t+1})\|^2 \right) - \delta
\tag{21}
$$

This correction is especially important in the early diffusion regime, where the signal-to-noise ratio is low and the posterior estimate is dominated by the offset. Under the EDM scheduler, where $\sigma_t^2 \in [0.002, 80]$, the transition variance $\sigma_{t-1|t}^2 = \sigma_t^2(1 - \sigma_{t-1}^2/\sigma_t^2)$ spans an extremely wide range, exacerbating this effect.

To make this more interpretable, we define $\delta$ in terms of $\pi$ and a new hyperparameter $t_0$, defined as the timestep at which the guidance scale reaches a reference value $\lambda_{\text{ref}} = 3$ under a purely linear model. This yields:

$$
\delta = \frac{1}{N_{\text{steps}}(1 - t_0)} \log \left( (1 - \pi) \cdot \frac{\lambda_{\text{ref}}}{\lambda_{\text{ref}} - 1} \right)
\tag{22}
$$

**Reparameterising the temperature $\tau$**

We now reparametrize $\tau$ using $t_0$ and $t_1$, where $t_1$ represents the timestep at which the additive Euclidean term matches the magnitude of the offset $\delta$. To estimate this, we compute the average contribution $\Delta$ of the Euclidean difference across sampled trajectories (see Fig. 6b). For the EDM2-XS model, we find $\Delta \approx 10$ around $t = 0.5$.

This gives:

**Algorithm 1** Feedback Guidance (FBG)

---

**Input:** Pre-trained conditional and unconditional network with prediction $\boldsymbol{\mu}_\theta(\boldsymbol{x}_t|c)$ and $\boldsymbol{\mu}_\theta(\boldsymbol{x}_t)$, mixing factor $\pi$, the two timestep hyperparameters $t_0$ and $t_1$ and a maximal guidance scale value $\lambda_{\max}$

Derive $\delta, \tau$ from $\pi, t_0, t_1$ and $p_{\min}$ from $\lambda_{\max}$ ⬛ Set hyperparameters (App. C.2)

$\boldsymbol{x}_T \sim \mathcal{N}(0, \boldsymbol{I})$ ⬛ Initialize state

$\log p(c|\boldsymbol{x}_T) = 0$ ⬛ Initialize posterior and guidance scale

$\boldsymbol{\lambda_T}(\boldsymbol{x}_T) = \frac{p(c|\boldsymbol{x}_T)}{p(c|\boldsymbol{x}_T)-(1-\pi)}$

**for** $t = T, \ldots, 1$ **do**

$\quad \boldsymbol{\mu}_{\theta,\text{guid}}(\boldsymbol{x}_t|c) = \boldsymbol{\mu}_\theta(\boldsymbol{x}_t) + \lambda_t(\boldsymbol{x}_t)\big(\boldsymbol{\mu}_\theta(\boldsymbol{x}_t|c) - \boldsymbol{\mu}_\theta(\boldsymbol{x}_t)\big)$ ⬛ Compute and mix scores

$\quad \boldsymbol{x}_{t-1} = \boldsymbol{\mu}_{\theta,\text{guid}}(\boldsymbol{x}_t|c) + \sigma_{t-1|t}\boldsymbol{z}$ with $\boldsymbol{z} \sim \mathcal{N}(0, \boldsymbol{I})$ ⬛ DDPM Step

$\quad \log p(c|\boldsymbol{x}_{t-1}) = \log p(c|\boldsymbol{x}_t)$ ⬛ Update the log posterior
$\quad \qquad + \frac{\tau}{2\sigma_{t-1|t}^2}\big(\|\boldsymbol{x}_{t-1} - \boldsymbol{\mu}_\theta(\boldsymbol{x}_t|c)\|^2 - \|\boldsymbol{x}_{t-1} - \boldsymbol{\mu}_\theta(\boldsymbol{x}_t)\|^2\big) - \delta$

$\quad \log p(c|\boldsymbol{x}_{t-1}) = \max\big(\log p(c|\boldsymbol{x}_{t-1}), \log p_{\min}\big)$ ⬛ Clamp the posterior

$\quad \lambda_t(\boldsymbol{x}_{t-1}) = \frac{p(c|\boldsymbol{x}_{t-1})}{1-p(c|\boldsymbol{x}_{t-1})}$ ⬛ Update the guidance scale

**end for**

---

$$\tau = \frac{2\tilde{\sigma}_{t_0}}{\Delta} \cdot \delta \tag{23}$$

While $\Delta$ is not a tunable hyperparameter, using it improves interpretability: $t_1$ now corresponds to the point at which guidance begins to decrease, marking the transition to effective conditional denoising.

**Summary**

By reparametrizing the abstract hyperparameters $\tau$ and $\delta$ in terms of normalized diffusion times $t_0$ and $t_1$, we provide a more intuitive interface for tuning Feedback Guidance. This improves both usability and interpretability, which we consider essential for practical deployment.

## D  Pseudocode of Feedback Guidance

Our Feedback Guidance procedure is summarized in Algorithm 1. At each denoising step, given a noisy state $\boldsymbol{x}_t$, we compute both the unconditional prediction $\boldsymbol{\mu}_\theta(\boldsymbol{x}_t)$ and the conditional prediction $\boldsymbol{\mu}_\theta(\boldsymbol{x}_t|c)$. These predictions are then combined using a guidance scale determined by the previously estimated posterior through eq.(8). The resulting mixture is used to predict the next, less noisy state $\boldsymbol{x}_{t-1}$. After this step, the posterior is updated based on the new state, and the process is repeated for a fixed number of iterations until a fully denoised image is produced.

Our code, compatible with the EDM2 repository Karras et al. [2024b,a], is provided to the reviewers as supplementary material.

## E  Stochastic sampling for Variance Exploding Diffusion Models

It is well known the forward process can be freely chosen. Two very standard cases are those of a *Variance Preserving* (VP) and that of a *Variance Exploding* (VE) forward process.

In the VP case, the information is progressively destroyed by both downscaling the features by a factor $\sqrt{\alpha_t}$ and adding normal noise with standard deviation $\sqrt{1-\alpha_t}$, i.e. $\boldsymbol{x}_{t+1} = \sqrt{\alpha_t}\boldsymbol{x}_t + \sqrt{1-\alpha_t}\boldsymbol{\epsilon}$ with $\boldsymbol{\epsilon} \sim \mathcal{N}(\boldsymbol{0}, \boldsymbol{I})$. This forward transition can alternatively be described by $\boldsymbol{x}_{t+1} \sim q(\boldsymbol{x}_{t+1}|\boldsymbol{x}_t)$ with $q(\boldsymbol{x}_{t+1}|\boldsymbol{x}_t) = \mathcal{N}(\sqrt{\alpha_t}\boldsymbol{x}_t, (1-\alpha_t)\boldsymbol{I})$. Thanks to a nice property of the Gaussian function, this Markov chain can be reparameterised as $\boldsymbol{x}_{t+1} \sim q(\boldsymbol{x}_{t+1}|\boldsymbol{x}_0)$ with $q(\boldsymbol{x}_{t+1}|\boldsymbol{x}_0) = \mathcal{N}(\sqrt{\bar{\alpha}_t}\boldsymbol{x}_0, (1-\bar{\alpha}_t)\boldsymbol{I})$ and $\bar{\alpha}_t = \prod_{s=0}^{t}\alpha_t$. This forward process can then be inverted according to $\boldsymbol{x}_{t-1} \sim p(\boldsymbol{x}_{t-1}|\boldsymbol{x}_t)$ with $p(\boldsymbol{x}_{t-1}|\boldsymbol{x}_t) = \mathcal{N}(\boldsymbol{\mu}_t, \tilde{\sigma}_t^2\boldsymbol{I})$. In this case, it is well known that $\boldsymbol{\mu} = \frac{1}{\sqrt{\alpha_t}}\big(\boldsymbol{x}_t - \frac{1-\alpha_t}{\sqrt{1-\bar{\alpha}_t}}\boldsymbol{\epsilon}_t\big)$ and $\tilde{\sigma}_t = \frac{1-\bar{\alpha}_{t-1}}{1-\bar{\alpha}_t}\beta_t$.

The case of VE is far less often described using the discrete markov chain framework, which is why we think it wise to derive the precise shape of $\boldsymbol{x}_{t-1} \sim p(\boldsymbol{x}_{t-1}|\boldsymbol{x}_t)$ with $p(\boldsymbol{x}_{t-1}|\boldsymbol{x}_t) = \mathcal{N}(\boldsymbol{\mu}_t, \tilde{\sigma}_t^2\boldsymbol{I})$. In the VE case, the forward process simply consists of adding gaussian noise of increasing scale

$x_{t+1} = x_t + \sqrt{\sigma_{t+1}^2 - \sigma_t^2}\epsilon$ with $\epsilon \sim \mathcal{N}(\mathbf{0}, \mathbf{I})$. Similarly to the VP case, the previous process can be reparameterise $x_{t+1} \sim q(x_{t+1}|x_0)$ with $q(x_{t+1}|x_0) = \mathcal{N}(x_0, \sigma_{t|0}^2 \mathbf{I})$ and $\sigma_{t|0}^2 = \sum s = 0^t \sigma_s^2$. To obtain the form of $p(x_{t-1}|x_t) = \mathcal{N}(\mu_t \sigma_{t-1|t}^2 \mathbf{I})$ we need to obtain $q(x_{t-1}|x_t, x_0)$ which is obtainable thanks to the conditioning on $x_0$. One finds:

$$p(x_{t-1}|x_t, x_0) = q(x_t|x_{t-1}, x_0)\frac{q(x_{t-1}|x_0)}{q(x_t|x_0)}$$

$$\propto \exp\Big(-\frac{1}{2}\Big[\frac{\|x_t - x_{t-1}\|^2}{\sigma_t^2 - \sigma_{t-1}^2} + \frac{\|x_{t-1} - x_0\|^2}{\sigma_{t-1}^2} + \frac{\|x_t - x_0\|^2}{\sigma_t^2}\Big]\Big)$$

$$= \exp\Big(-\frac{1}{2}\Big[\Big(\frac{1}{\sigma_t^2 - \sigma_{t-1}^2} + \frac{1}{\sigma_{t-1}^2}\Big)\|x_{t-1}\|^2 \qquad (24)$$

$$- 2\Big(\frac{1}{\sigma_t^2 - \sigma_{t-1}^2}x_t + \frac{1}{\sigma_{t-1}^2}x_0\Big)x_{t-1} + c(x_t, x_0)\Big]\Big)$$

$$\propto \exp\Big(-\frac{\|x_{t-1} - \mu_t\|^2}{2\sigma_{t-1|t}^2}\Big)$$

Rewriting the clean data points $x_0$ using the ground truth noise $\epsilon_t$ through $x_0 = x_t - \sigma_t \epsilon_t$. This is done because this is precisely how our denoising network will be parametrized. Using this, one recognizes:

$$\sigma_{t-1|t}^2 = (\sigma_t^2 - \sigma_{t-1}^2)\frac{\sigma_{t-1}^2}{\sigma_t^2}$$

$$\mu_t = x_t - \Big(1 - \frac{\sigma_{t-1}^2}{\sigma_t^2}\Big)\sigma_t \epsilon_t \qquad (25)$$

Or alternatively using the score function, i.e. $\epsilon_t = \sigma_t \nabla_x \log p_t$, we have:

$$\mu_t = x_t - \big(\sigma_t^2 - \sigma_{t-1}^2\big)\nabla_x \log p_t \qquad (26)$$

Or equivalently, the very intuitive equation:

$$\nabla_x \log p_t = \frac{x_t - \mu_t}{\sigma_t^2 - \sigma_{t-1}^2} \qquad (27)$$

# F  Additional results and ablations

In this appendix all the additional ablations and obtained results are provided and described in more detail.

## F.1  Detailed description of class-conditional experiments

First and foremost the hyperparameters of the different guidance schemes that minimize the FID or $FD_{DinoV2}$ are provided in Table 3. It should also be noted that CFG++ Chung et al. [2025] was originally not analyzed in the context of class-conditional image generation such as we do on Imagenet, explaining the performance observed in Table 1. On the other hand, the guidance weight-schedulers introduces by Wang et al. [2024], were only verified using the FID-metric, much less sensible to late stage guidance than the $FD_{DinoV2}$, explaining the underperformance in that regime. For CFG Ho and Salimans [2021] a sweep over the guidance scale is performed at a resolution of $\lambda = 0.1$.

To compare our method with adaptive CFG weight schedulers, we follow Wang et al. [2024] and benchmark against their best-performing variant, the linearly increasing scheduler, which we denote as LinCFG. The reported guidance scale corresponds to the trajectory-averaged value, and we sweep over $\lambda$ with a resolution of $0.1$. As expected, the optimal scales of LinCFG strongly correlate with those of standard CFG.

For CFG++ Chung et al. [2025], we perform a sweep with a finer resolution of $\lambda = 0.025$. Unlike other guidance schemes, CFG++ constrains $\lambda \in [0, 1]$, since it modifies not only the score function prediction but also the coupling between forward and reverse diffusion processes. This design

| Guidance scheme | CFG | LinCFG | CFG++ | LIG | | | FBG$_{\text{pure}}$ | | | FBG$_{\text{LIG}}$ | |
|---|---|---|---|---|---|---|---|---|---|---|---|
| | $\lambda$ | $\lambda$ | $\lambda$ | $\lambda$ | $\sigma_{\max}$ | $\sigma_{\min}$ | $\pi$ | $\sigma_{t_0}$ | $\sigma_{t_1}$ | $\lambda$ | $\sigma_{t_1}$ |
| Stoch. (FID) | 1.4 | 1.5 | / | 2.8 | 1.6 | 0.15 | 0.999 | 1.10 | 0.56 | 1.4 | 4.64 |
| PFODE (FID) | 1.4 | 1.5 | 0.35 | 2.2 | 2.9 | 0.41 | 0.999 | 1.61 | 0.60 | 2.6 | 2.7 |
| Stoch. (FD$_{\text{DinoV2}}$) | 2.1 | 2.1 | / | 2.9 | 6.8 | 0.48 | 0.999 | 4.07 | 1.29 | 2.6 | 2.34 |
| PFODE (FD$_{\text{DinoV2}}$) | 2.3 | 2.2 | 0.6 | 2.8 | 16.6 | 0.80 | 0.999 | 6.46 | 1.17 | 1.6 | 1.61 |

Table 3: Optimal hyperparameters for different sampling approaches. To facilitate the comparison between the schemes we follow the nomenclature introduced of the EDM framework Karras et al. [2022] and refer to the noise levels $\sigma_{t_0}$, $\sigma_{t_1}$ instead of normalized diffusion times $t_0$ and $t_1$ for FBG. For FBG$_{\text{LIG}}$ the unspecified parameters ($\pi$, $t_0$, $\sigma_{\max}$ and $\sigma_{\min}$) are left unaltered from the separately optimised methods.

precludes its use with purely stochastic DDPM samplers, which do not rely on forward noising during denoising. In essence, CFG++ is conceptually distinct from conventional guidance schemes and could, in principle, be combined with methods such as LIG or other weight schedulers. We include this benchmark to provide a more comprehensive comparison with alternative approaches proposed in the literature.

For LIG a joint sweep over the guidance scale and the starting point of guidance is done. The sweep over the guidance scale is performed at a resolution of $\lambda = 0.25$ and the starting point $\sigma_{\max}$ is chosen at the discretised step values. The influence of the end point of guidance is not analysed, instead a low value of $\sigma_{\min} = 0.28$ is chosen. As higlighted in the work in which the method is introduced, increasing $\sigma_{\min}$ leaves the FID unaltered and is simply beneficial from a computational point of view Kynkäänniemi et al. [2024].

For FBG a joint sweep over $t_0$ and $t_0 - t_1$ is performed defined as the normalised diffusion times corresponding with the discrete step values. For instance in the case of stochastic sampling where 64 sampling steps are used we perform a sweep at a resolution of $t_0 = 1/64 \simeq 0.0156$. We prefer to define $t_1$ as a function of $t_0$, as their difference gives an estimate for how large the guidance interval is. This is then repeated for three values of $\pi = 0.999, 0.9999, 0.99999$. These values were chosen after a preliminary finetuning by hand. We find that although the guidance profiles do differ slightly between different choices of $\pi$, mainly being sharper as $\pi$ increases, the optimal FID/FD$_{\text{DinoV2}}$ remain very similar. Due to this we choose to focus our limited resources on a proper ablation at a fixed value of $\pi = 0.9999$. To illustrate the weak dependence of FBG on values of $\pi$ a sweep is performed using the optmal values for $t_0$ and $t_1$ given in table 3. The results are shown in Fig. 9, where it can be seen that both FID and FD$_{\text{DinoV2}}$ values remain fairly constant over a wide range of $\pi$-values.

It should also be noted that at some point in the research process we tried adding a late start to the offset parameter, i.e. to set $\delta = 0$ for $t > t_{\text{start-offset}}$, and slightly modify the way we parameterise $\delta$ as a function of $t_0$ such that its interpretation remains true. This however did not significantly modify the performance of the approach, so this research track was dropped to avoid any unnecessary convolutions. For all methods we find that the FD$_{\text{DinoV2}}$-optimal hyperparameter values result in a much higher amount of guidance than the FID-optimized values, hinting that the metrics are not sensitive to the same features Stein et al. [2023]. This fact highlights that only providing one of the two metrics when benchmarking a new approach might not provide the full story.

Figure 8 is the corresponding figure to Fig. 3 in the main body, but for the FID-optimised stochastic sampling, rather than for FD$_{\text{DinoV2}}$.

## F.2 Detailed description of T2I experiments

First and foremost the hyperparameters of the three guidance schemes that minimize the FID or FD$_{\text{DinoV2}}$ are provided in Table 4.

For CFG Ho and Salimans [2021] a sweep over the guidance scale is performed at a resolution of $\lambda = 0.25$.

For LIG Kynkäänniemi et al. [2024], we follow the recommendations of Kynkäänniemi et al. [2024], and perform a joint sweep over the guidance scale and $\sigma_{\max}$ at a resolution of $\lambda = 0.25$. Thereafter, $\sigma_{\min}$ is optimized. As expected, we find that the FD$_{\text{DinoV2}}$ optimal interval is much larger and earlier. For FBG, we perform a sweep over $t_0$ and $t_0 - t_1$ at a resolution of $0.05$. We then perform a sweep over $\pi$ on a logarithmic scale, similar to that used in Fig. 9.

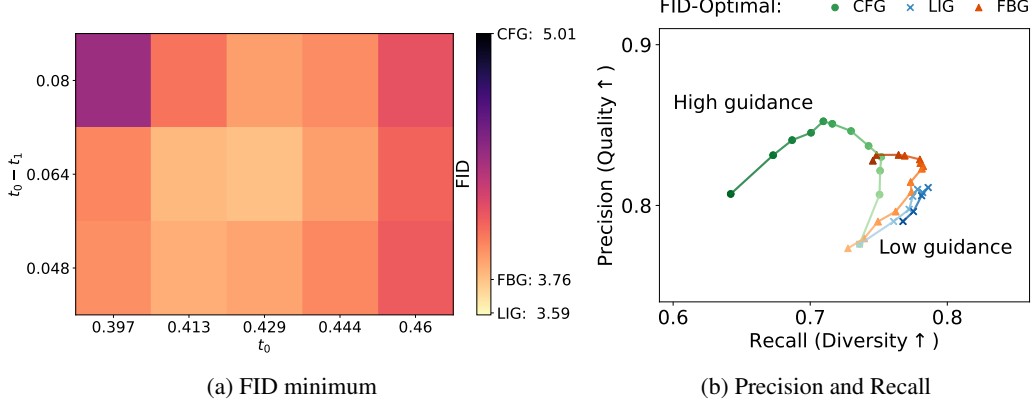

(a) FID minimum

(b) Precision and Recall

Figure 8: Grid search over the two hyperparameters $t_0$ and $t_1$. The FID is shown calibrated within the best performing value using LIG and CFG.

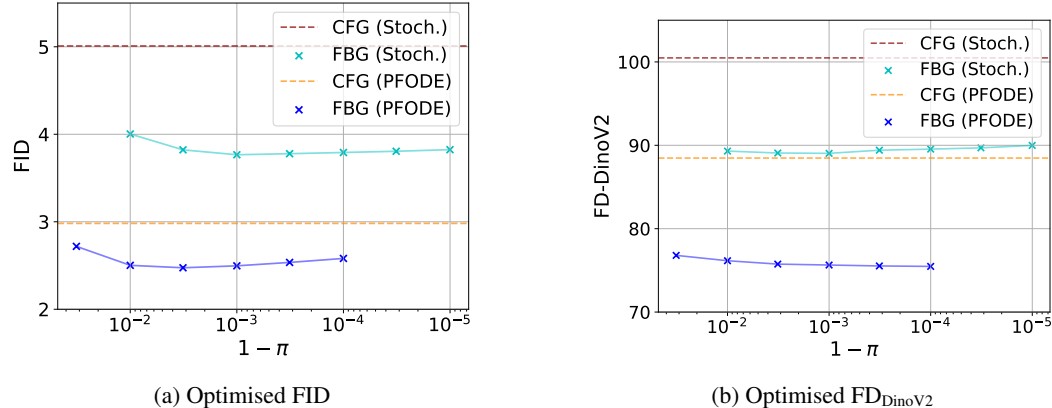

(a) Optimised FID

(b) Optimised FD$_{\text{DinoV2}}$

Figure 9: Illustration of the weak $\pi$ dependence of FBG when parametrised using $t_0$ and $t_1$. The sweep over $\pi$ is performed at the optimal values for $t_0$ and $t_1$ given to in Table 3.

| Guidance scheme | CFG | LinCFG | LIG | | | FBG$_{\text{pure}}$ (ours) | | | FBG$_{\text{CFG}}$ (ours) | | |
| --- | --- | --- | --- | --- | --- | --- | --- | --- | --- | --- | --- |
| | $\lambda$ | $\lambda$ | $\lambda$ | $\sigma_{\max}$ | $\sigma_{\min}$ | $\pi$ | $t_0$ | $t_1$ | $\lambda$ | $t_0$ | $t_1$ |
| Stoch. (FID) | 2.25 | 3.25 | 4.0 | 2.4 | 0.08 | 0.9 | 0.55 | 0.4 | 1.0 | 0.55 | 0.4 |
| Stoch. (FD$_{\text{DinoV2}}$) | 2.5 | 3.0 | 4.0 | 3.94 | 0 | 0.9 | 0.65 | 0.45 | 2.0 | 0.5 | 0.375 |

Table 4: Optimal hyperparameters when optimizing for FID and FD$_{\text{DinoV2}}$ on MS-COCO using Stable DIffusion 2. For FBG$_{\text{CFG}}$, the hyperparameter $\pi$ is left unaltered from FBG$_{\text{pure}}$. We also note that in the context of FID optimisation, FBG$_{\text{CFG}}$ reduces to FBG$_{\text{pure}}$, that is that additional classifier guidance on top of FBG$_{\text{pure}}$ is harmful, which is not the case when optimizing using the FD$_{\text{DinoV2}}$.

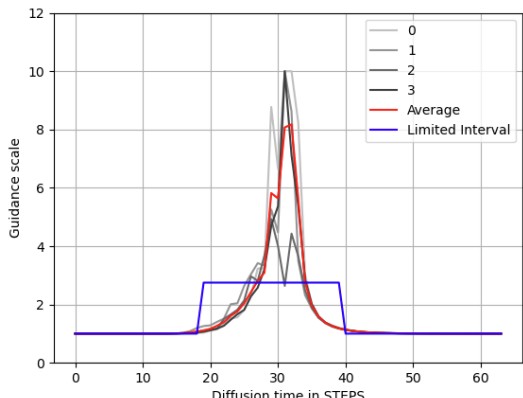

Figure 10: Comparing the guidance scale predicted by FBG and that of LIG in the $\text{FD}_{\text{DinoV2}}$ optimized setting. Despite possessing similarities the two seem to operate in different regimes.

We would also here like to emphasize that we believe the MS-COCO benchmark to be suboptimal when comparing different guidance schemes. This is because the prompts of MS-COCO Lin et al. [2014] are quite uniform and fairly standard, which precisely corresponds to settings when guidance is not needed as much. This explains why the optimized guidance scales are much smaller than the ones typically used for sampling complex prompts, such as the ones used in our handcrafted dataset as shown in Fig. 4. This benchmark however remains the standard used in the literature, which is why we choose to report it here.

### F.3   Combined guidance schemes

The proposed Feedback Guidance scheme can be easily combined with other preexisting guidance schemes such as Classifier-Free Guidance Ho and Salimans [2021] or Limited Interval Guidance Kynkäänniemi et al. [2024].
In the context of Text-To-Image we observe that adding a base level of CFG can help to retrieve the low frequency features of an image, such as sharp colors, without drastically harming the diversity. In the context of Imagenet generation using EDM2-XS, we find that using $\text{FBG}_{\text{LIG}}$ which combines $\text{FBG}_{\text{pure}}$ with LIG, is optimal. Preliminary results indicate that joint methods easily outperform their parts. That such results are obtained in this context should not surprise the reader, both method despite having some similarities, behave very differently as illustrated in Fig. 10. To simplify hyperparameter tuning of $\text{FBG}_{\text{LIG}}$, we suggest to use the optimal time interval parameters of LIG with slightly less guidance and to slightly reduce $t_1 - t_0$ for $\text{FBG}_{\text{pure}}$. In essence, both of these subtle changes are responsible for less guidance of the respective schemes, which makes sense as the two are later on combined.
The optimal values for the sweep over $\lambda_{\text{LIG}}$ and $t_1$ are given in Table 3. The other parameters are chosen the same as the separately optimized methods. We do not exclude the possibility that a full grid-search over the entire joint hyperparameter space might yield better trade-offs between the two guidance schemes.

An advantage of the error assumption model proposed is that it allows for a very flexible view of guidance. For instance, for the $\text{FBG}_{\text{CFG}}$ approach in fact simply coresponds to assuming that the true conditional distribution can be rewritten as:

$$p_t(\boldsymbol{x}_t|c) = \frac{1}{\pi}\big(p_{t,\theta}(\boldsymbol{x}_t|c) - (1-\pi)p_\theta(\boldsymbol{x}_t)\big)\frac{p_{t,\theta}(\boldsymbol{x}_t|c)^{\lambda-1}}{p_{t,\theta}(\boldsymbol{x}_t)^{\lambda-1}} \tag{28}$$

This implies that the learned distribution satisfies the following algebraic equation:

$$p_{t,\theta}(\boldsymbol{x}_t|c)^\lambda - (1-\pi)p_{t,\theta}(\boldsymbol{x}_t|c)^{\lambda-1}p_t(\boldsymbol{x}_t) = \pi p_t(\boldsymbol{x}_t|c)p_t(\boldsymbol{x}_t)^{\lambda-1} \tag{29}$$

Similarly, $\text{FBG}_{\text{LIG}}$ assumes the same form of error with as only distinction that $\lambda$ becomes a time dependent function that is equal to one outside the guidance interval specified by $\sigma_{\min}$ and $\sigma_{\max}$.

## F.4 Illustrative samples (EDM2-XS)

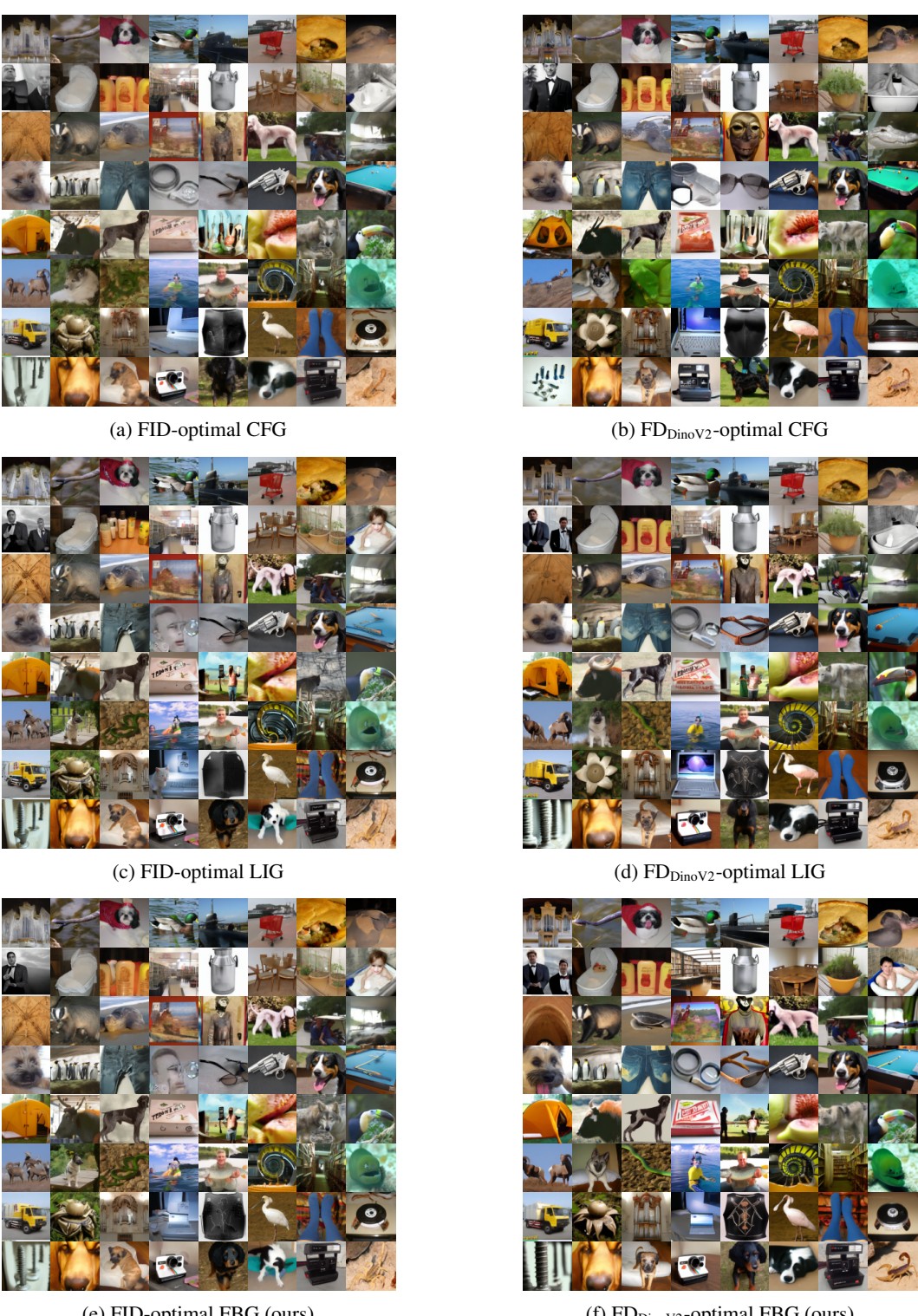

(a) FID-optimal CFG

(b) FD$_{DinoV2}$-optimal CFG

(c) FID-optimal LIG

(d) FD$_{DinoV2}$-optimal LIG

(e) FID-optimal FBG (ours)

(f) FD$_{DinoV2}$-optimal FBG (ours)

Figure 11: Grids containing samples generated using the different guidance schemes CFG, LIG and FBG (ours). Results displayed under the same seed and when he hyperparameters are optimised for both FID and FD$_{DinoV2}$-optimal performance.

### F.5 Illustrative samples (T2I)

To facilitate comparison of our newly introduced FBGpure and FBG_{CFG} schemes with CFG, we provide illustrative samples. Prompts are drawn from our curated dataset spanning four difficulty levels (memorized < basic < intermediate < hard; see Appendix G). For each level, two prompts are randomly selected, and we display four samples generated with: (i) the conditional model Rombach et al. [2022], (ii) CFG with guidance scale 3.5 Ho and Salimans [2021], (iii) FBGpure with $(\pi = 0.9, t_0 = 0.75, t_1 = 0.5)$, and (iv) FBG_{CFG} with an additional fixed scale 1.5.

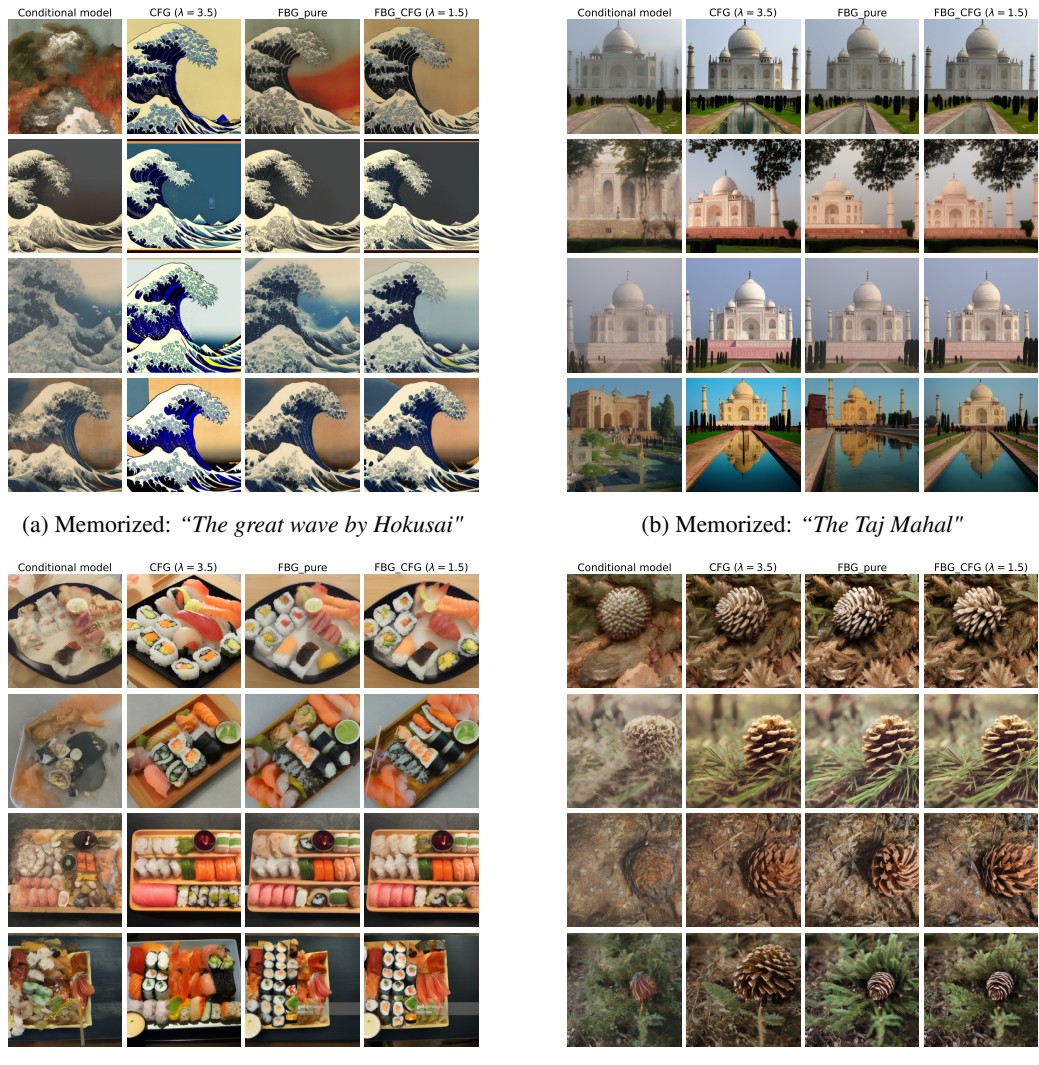

(a) Memorized: *"The great wave by Hokusai"*

(b) Memorized: *"The Taj Mahal"*

(c) Easy: *"A sushi platter"*

(d) Easy: *"A pinecone resting on a forest floor"*

Figure 12: Different samples for randomly selected prompts of memorized and easy prompts. The used prompts are written underneath the images.

For memorized/easy prompts, visible in Figure 12, CFG images often exhibit oversaturated colors and overly smooth textures, whereas FBG variants largely avoid these artifacts by deactivating guidance when the conditional prediction is already accurate.

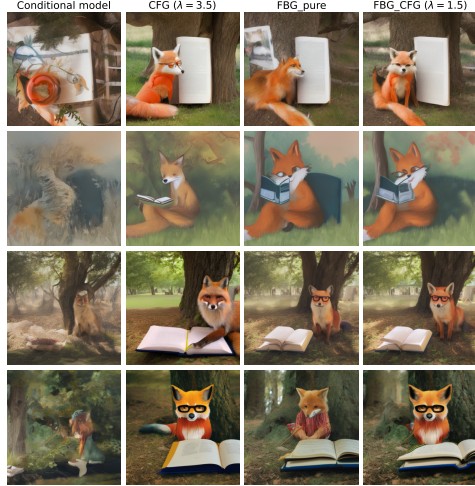

(a) Intermediate: *"A fox wearing reading glasses sitting under a tree with an open book"*

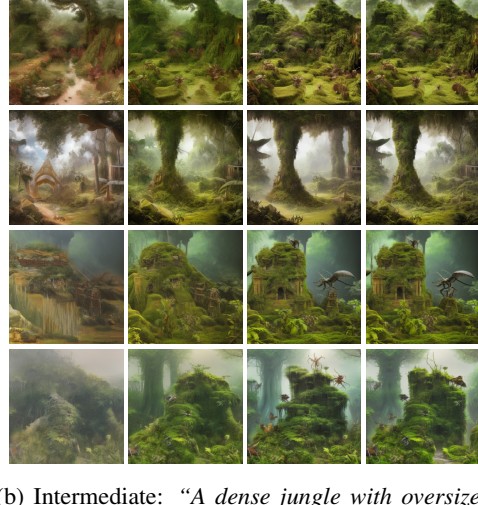

(b) Intermediate: *"A dense jungle with oversized insects and ruins covered in moss"*

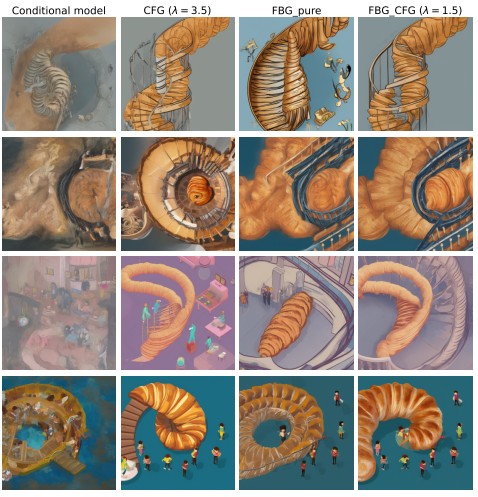

(c) Hard: *"A croissant slowly unrolling itself into a spiral staircase, with tiny chefs walking up each layer, in detailed isometric art style"*

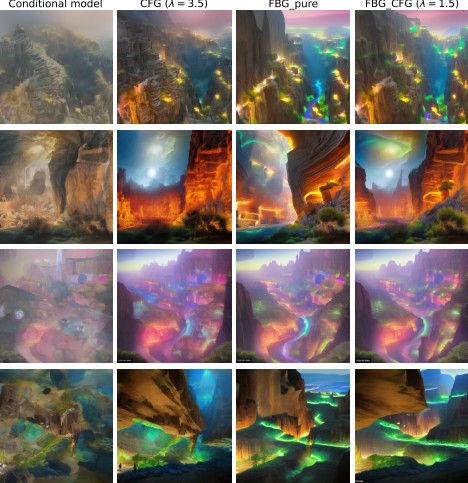

(d) Hard: *"A city built inside a giant canyon where each layer of rock houses a different civilization, all lit by bioluminescent flora"*

Figure 13: Different samples for randomly selected prompts of intermediate and hard prompts. The used prompts are written underneath the images.

Harder prompts show the strength of FBG, e.g., capturing bioluminescent lights in Fig. 13d, as its dynamic scale allocates stronger guidance only when needed. Such challenging prompts are far more informative for comparing guidance schemes than simpler MS-COCO captions Lin et al. [2014]. It should be however noted that both schemes struggle with following all the details present in the prompts.

Next, we present qualitative comparisons between images generated with CFG and FBG$_{\text{CFG}}$, along with the corresponding dynamic guidance scales across different prompt difficulty levels. As discussed in the main paper, the guidance scale in FBG increases with prompt complexity.
For memorized prompts (e.g., Figures 14 and 15), the dynamic guidance scale remains near one, preserving sample diversity. In contrast, CFG tends to overemphasize prompt-specific details, leading to oversaturated features such as excessively bright colors.
For challenging prompts (e.g., Figures 18 and 19), FBG applies a much higher guidance scale, successfully enforcing the generation of key prompt-specific features, such as the phoenix in Figure 19, that are underrepresented in CFG outputs.

In both cases, the fixed guidance scale used by CFG is suboptimal: it is too strong for memorized prompts and too weak for difficult ones. These results reinforce our central claim that guidance should not rely on a global, fixed scale, but instead adapt dynamically, activating only when needed to enhance fidelity, while remaining inactive to preserve diversity when the conditional model already performs well.

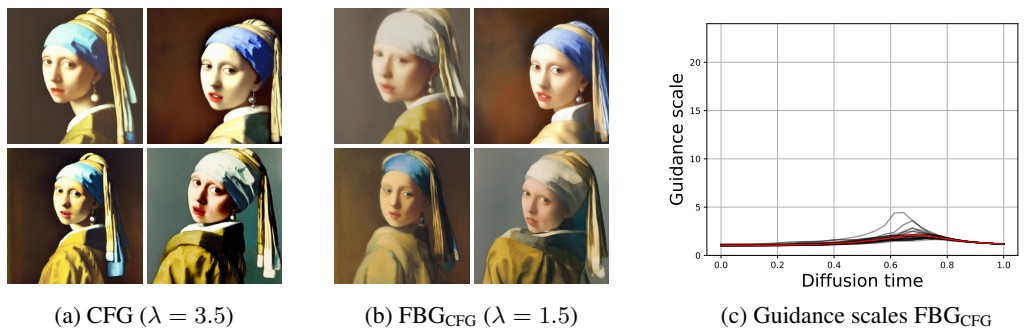

(a) CFG ($\lambda = 3.5$)     (b) FBG$_\text{CFG}$ ($\lambda = 1.5$)     (c) Guidance scales FBG$_\text{CFG}$

Figure 14: Different samples for the memorized prompt: *"Girl with pearl by Vermeer"*

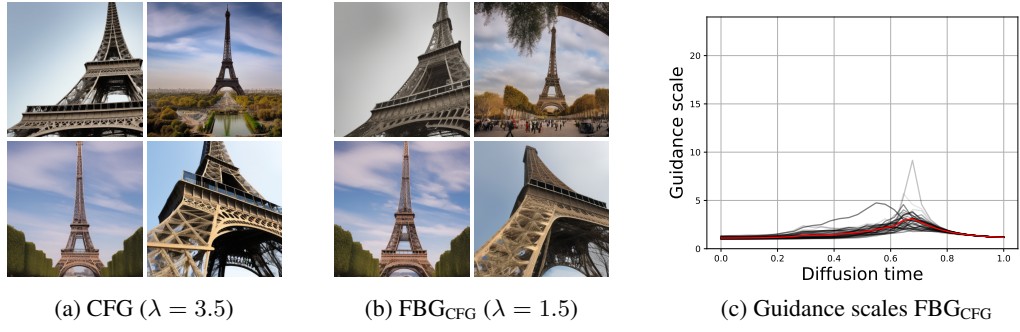

(a) CFG ($\lambda = 3.5$)     (b) FBG$_\text{CFG}$ ($\lambda = 1.5$)     (c) Guidance scales FBG$_\text{CFG}$

Figure 15: Different samples for the memorized prompt: *"The Eiffel Tower in Paris"*

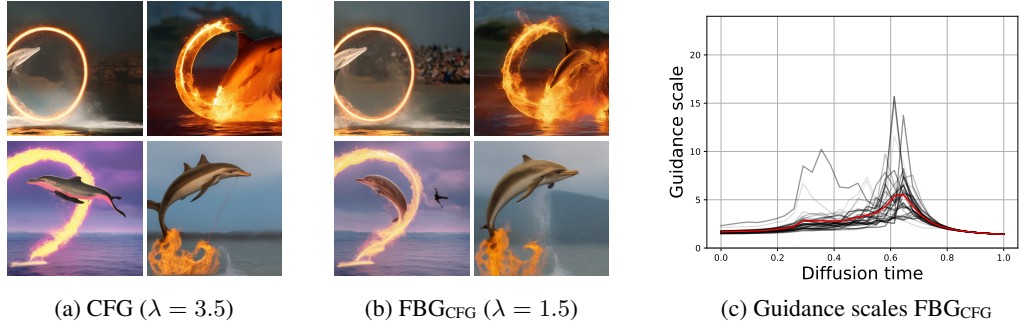

(a) CFG ($\lambda = 3.5$)     (b) FBG$_\text{CFG}$ ($\lambda = 1.5$)     (c) Guidance scales FBG$_\text{CFG}$

Figure 16: Different samples for the intermediate prompt: *"A dolphin jumping through a hoop made of fire"*

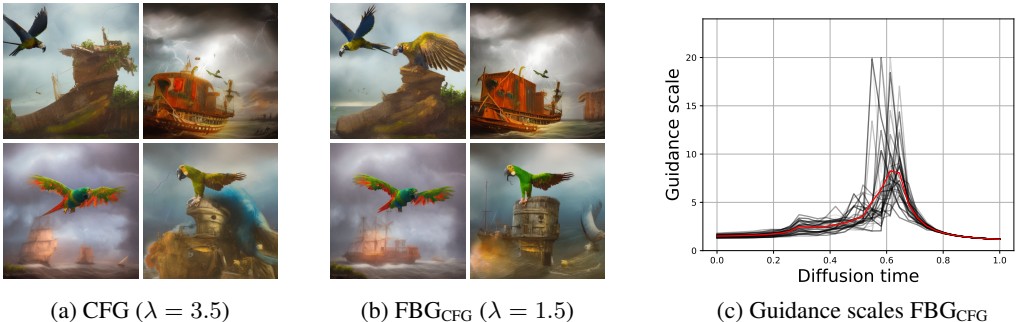

(a) CFG ($\lambda = 3.5$)    (b) FBG$_{\text{CFG}}$ ($\lambda = 1.5$)    (c) Guidance scales FBG$_{\text{CFG}}$

Figure 17: Different samples for the intermediate prompt: *"A parrot with steampunk goggles flying through a thunderstorm above a 19th-century shipwreck, in dramatic oil painting style"*

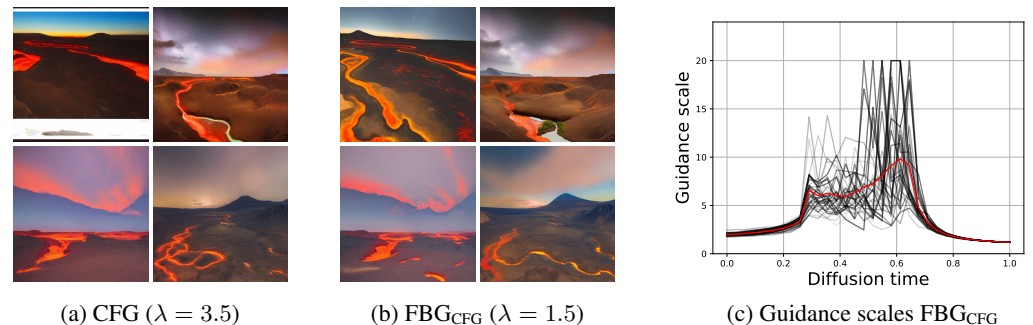

(a) CFG ($\lambda = 3.5$)    (b) FBG$_{\text{CFG}}$ ($\lambda = 1.5$)    (c) Guidance scales FBG$_{\text{CFG}}$

Figure 18: Different samples for the intermeiate prompt: *"A volcanic landscape with rivers of lava flowing under a starry sky"*

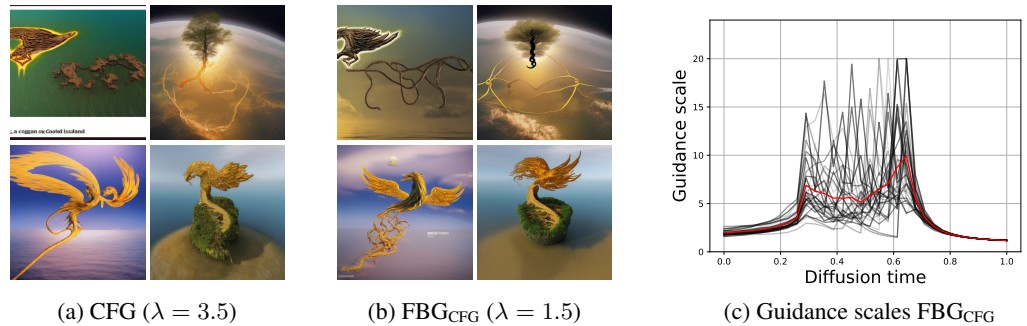

(a) CFG ($\lambda = 3.5$)    (b) FBG$_{\text{CFG}}$ ($\lambda = 1.5$)    (c) Guidance scales FBG$_{\text{CFG}}$

Figure 19: Different samples for the intermeiate prompt: *"A floating island chained to the earth by golden vines, casting a shadow shaped like a phoenix over the ocean below"*

| Difficulty | Category | Prompt |
|---|---|---|
| Memorized | Artwork | The starry night by Van Gogh |
| Memorized | Brand | The Nike brand logo |
| Memorized | Location | A photo of the Taj Mahal in India |
| Basic | Animal | An orange cat sitting on a couch |
| Basic | Food | A pizza slice on a white plate |
| Basic | Nature | A bird nest with three blue eggs |
| Intermediate | Animal | A dolphin jumping through a hoop made of fire |
| Intermediate | Food | A floating sushi platter arranged in the shape of a koi fish, hovering over a pond |
| Intermediate | Nature | A desert landscape with an abandoned train half-buried in the sand |
| Very hard | Animal | A parrot with steampunk goggles flying through a thunderstorm above a 19th-century shipwreck, in dramatic oil painting style |
| Very hard | Food | A glass teapot filled with herbal tea where each herb leaf is shaped like a different mythical creature, photographed on white marble |
| Very hard | Nature | A city built inside a giant canyon where each layer of rock houses a different civilization, all lit by bioluminescent flora |

Table 5: Example prompts from the dataset, sorted by difficulty and category

# G    Prompt dataset

To analyse the sensitivity of our dynamic guidance scale to the complexity of the given prompts, a small scale prompt dataset is introduced. It contains 60 prompts of 4 difficulty levels: memorized, basic, intermediate and hard. Each complexity level is divided into 3 categories/topics containing 5 prompts each.
For the memorized prompts we use: well known artworks, brands and locations.
For the other three we use: animal, food and nature images.
The prompts themselves are generated using ChatGPT and further minimally modified to make the prompt easier to verify. The main difference between *basic* and *intermediate* prompts is that the latter contain highly unlikely combinations (such as "A giraffe playing basketball on rollerskates") that the model has most likely not seen (or only rarely) as such in the training data. The main difference between *intermediate* and *hard* prompts is mainly the amount of details contained in the prompt. The more details such as colours, numbers or different elements are added, the more unlikely it becomes that the conditional model will be to satisfy all prerequisites on its own.
The dataset is available in the official repository at this link. Examples of each difficulty level and each category are given in Table 5.

# H    Used resources and LLM use

For the stochastic sampler all experiments are run using on a NVIDIA Tesla V100-SXM3-32GB GPU. Generating 50k images in batches of 64 using the EDM2-XS model Karras et al. [2024b,a], as required for a valid FID benchmark Heusel et al. [2017], Stein et al. [2023], takes 7h30 on such a node. For the 2nd-order Heun sampler of the PFODE Karras et al. [2022, 2024b] a NVIDIA GeForce RTX 4090 GPU is used. Generating 50k images in batches of 64 using the EDM2-XS model Karras et al. [2024b,a], as required for a valid FID benchmark Heusel et al. [2017], Stein et al. [2023], takes 8h on such a node. For the T2I results using Stable diffusion 2Rombach et al. [2022], we rely on an NVIDIA Tesla V100-SXM3-32GB GPU. The experiments performed only require the generation of 3k images per hyperparameter setting, which takes around 2h on such a node.

During the writing of this document, publicly available LLMs of different sources were used to rewrite, or polish, existing text. Typically, a first draft was written by one of the authors and then polished after comments from the others, thereafter the text was, in some cases, condensed or slightly modified, on a paragraph level. It is our belief that by solely correcting the document with an LLM on a paragraph level, our original ideas as well as the intended flow of the paper remains closest to ours. The propositions of the LLMs were only accepted when in full alignment with the handwritten text, and were otherwise discarded.

