# OpenReview forum: "Feedback Guidance of Diffusion Models"
_NeurIPS.cc/2025/Conference — NeurIPS 2025 poster_

### Official Review · Reviewer_WucL · 2025-06-16

**Clarity:** 2
**Significance:** 2
**Originality:** 2
**Rating:** 4
**Confidence:** 2

**Summary:**

This paper introduces Feedback Guidance (FBG), a new method for dynamically adjusting the guidance scale in conditional diffusion models. Unlike traditional Classifier-Free Guidance (CFG), which uses a fixed guidance scale throughout the sampling process, FBG adapts the guidance based on the current state of the sample and the progress in the denoising process.

**Questions:**

None

**Ethical Concerns:**

["NO or VERY MINOR ethics concerns only"]

**Final Justification:**

The rebuttal has partially addressed my concerns. I still find the experiments to be not entirely convincing, but I remain positive about the well-motivated approach and still lean a bit towards acceptance.

**Limitations:**

-Additive error model is a simplification.
-Posterior estimation has a self-reference bias, mitigated using a correction term (δ).
-Evaluation limited to small models (EDM2-XS); large-scale testing pending.

**Quality:**

2

**Strengths And Weaknesses:**

Strengths
-Principled Theoretical Framework: Introduces a state- and time-dependent guidance mechanism derived from first principles, contrasting with heuristic-based methods like CFG or LIG.
-Improved Quality-Diversity Trade-off:Feedback Guidance outperforms Classifier-Free Guidance (CFG) in both fidelity (FID) and diversity (Recall), and performs competitively with Limited Interval Guidance (LIG).
-Adaptivity to Sample and Prompt: Guidance dynamically adjusts based on the current sample state and prompt complexity, enabling fine-grained control over generation without manual tuning.
-Trajectory-Specific Behavior: Unlike static methods, FBG can vary guidance per sample trajectory, which helps avoid over- or under-guiding individual outputs.
-Empirical Validation Across Tasks: Results are shown for both class-conditional generation (ImageNet) and text-to-image generation (Stable Diffusion 2), demonstrating broad applicability.
-Extensible and Modular: Can be easily combined with other methods (e.g., CFG + FBG or LIG + FBG), and integrates well with existing diffusion frameworks.

Weaknesses
-Computational Overhead: Estimating posterior likelihoods during sampling introduces additional computation, which may slow down inference compared to simpler fixed-guidance methods.
-Posterior Estimation Bias: The approach relies on self-evaluation of the conditional model, which can lead to biased posterior estimates. This is partly mitigated by a correction term, but still a limitation.

---

> ### Author Rebuttal · Authors · 2025-07-31
>
> First and foremost we would like to thank the reviewer for their detailed yet concise feedback. We appreciate that you recognize that our method is supported by strong mathematical foundations as well as the fact that it brings interesting insights in the context of prompt/trajectory-specific behavior.
>
> ## Response to Weakness #1: Computational overhead
>
> One of the key advantages of the iterative posterior estimation approach is that it comes at negligible additional cost as it solely relies on precomputed quantities, namely the conditional and unconditional score functions already required for the denoising itself. To further emphasize this point we have added a sentence in the main document in section 3.3 mentioning:
> > “A crucial advantage of computing the posterior using the scheme describe above is that it causes negligible computational overhead as all required quantities, in particular $\mathbf{\mu}_c(\mathbf{x}_t|c)$ and $\mathbf{\mu}(\mathbf{x}_t)$, are already computed.”
>
> ## Response to Weakness #2: Posterior estimation parametrisation
>
> The specific choice of parametrizing the posterior through the additional bias parameter $\delta$ is described in detail in appendix B2. While it is true that we could have made other choices, we believe that the choice of a linear transformation is well defended in the paper. It should also be pointed out, that even though this is the simplest possible choice, it still manages to lead to a method outperforming standard baselines, which we believe to be a strong demonstration of the potential of our approach. In the future, we are looking forward to seeing more work analyzing the potential of different parametrizations of the posterior.
>
> ## Response to Weakness #3: Limited evaluation
>
> Finally, we wish to address the reviewer's concern that the small-scale experiments may not be sufficient for assessing the method's significance. We agree that strengthening the paper in this regard would significantly improve its quality and impact.
>
> ### Extension from small EDM2-XS to larger EDM2-L
> To address this, during the rebuttal period we conducted the same analysis previously performed with the EDM2-XS model using the much larger EDM2-L model, providing more comprehensive and qualitative insights. The initial results, computed on 10k generated images, are presented in Table 1 below. The extension towards 50k generated images, as well as the addition of the Precision and Recall metrics, is under way but will not be ready in time for the initial rebuttal.
> We can observe that the results on EDM2-L are more similar for the different methods, compared to the small model. This is because the EDM2-L model is far more qualitative and requires much less guidance, a result expected from our interpretation of guidance as an error correcting scheme. As a consequence, all guidance approaches are more similar to each other.
>
> **Table 1: Performance comparison on EDM2-L (10k images)**
>
> | Method | FID | FD$_{\text{DinoV2}}$
> |--------|---------|----------------------|
> | CFG | 8.15 | 119.70 |
> | LIG | 7.78 | **112.68** |
> | FBG |**7.75** | 114.90 |
>
> ### Additional T2I results with Stable Diffusion 2 on MS-COCO 30k
> Similarly, to further support our findings in the T2I setting, we evaluated the FID, FD$_{\text{DinoV2}}$  alongside the average CLIP and Aesthetic Scores for different guidance schemes (CFG, LIG, and our proposed FBG) using Stable Diffusion 2 on the standard MS-COCO 30k dataset. These results, currently limited to a subset of 3k prompts, are reported in Table 2. Results on all 30k prompts will become available next week.
> The MS-COCO30k dataset contains prompts that originate from captioned images, which all remain fairly standard. We expect our approach to work even better in context where the prompts are much more varied. To remain consistent with the literature we however still chose to evaluate on this standard benchmark, on which FBG still slightly outperforms alternative approaches.
>
> **Table 2: Quantitative evaluation of T2I on MS-COCO benchmark (3k images)**
>
> | Method | FID | FD$_{\text{DinoV2}}$ | CLIP-Score| Aesthetic Score|
> |--------|---------|----------------------|-----------|--------|
> | CFG | 19.64 | 53.55 | 0.310 | 5.650 |
> | LIG | 18.81 | 53.55$^*$| 0.311 | 5.742 |
> | FBG |**18.62** | **52.93** | 0.311 |**5.753**|
>
> $^*$In this setting, the best LIG interval corresponded to applying guidance on the whole trajectory.
>
> We hope that these results help convince the reviewers of the more general validity of the method, as well as of our commitment to incorporate their valid critique to improve the current submission.
>
> ## Summary
> We appreciate the reviewer's concerns and questions, and are convinced our responses as well as the additional experimental results have the potential to improve the manuscript considerably. We hope the reviewer reassesses the manuscript in light of this rebuttal, to see whether it warrants increasing their original score.

---

> > ### Comment · Reviewer_WucL · 2025-08-07
> >
> > Thanks to the authors for responding to my comments.
> >
> > The additional experimental results are somewhat helpful but not very convincing, as they exhibit only marginal improvements and have not yet been extended to a large number of generated images and prompts. Nevertheless, I still appreciate the principled theoretical framework and believe that this well-motivated approach is a good contribution. Thus, I have decided to maintain my original rating.

---

> > > ### Author Response · Authors · 2025-08-07
> > >
> > > ### Thank you for your reply!
> > >
> > > We sincerely thank the reviewer for their thoughtful comments and for maintaining their positive assessment of our work.
> > >
> > > We are currently running the grid search on the standard number of generated images (50k and 30k for EDM2 and MS-COCO), though completing these experiments requires a bit more time.
> > >
> > > We appreciate your understanding and your recognition of the theoretical motivation behind our approach, which we see as a central strength of the work.

---

### Official Review · Reviewer_maqm · 2025-06-28

**Clarity:** 2
**Significance:** 3
**Originality:** 3
**Rating:** 3
**Confidence:** 2

**Summary:**

This paper study the classifier-free guidance (CFG) in diffusion model. The traditional guidance scheme harm the diversty and memory-inefficient due to the constant guidance design. To tackle this problem, the authors propose a novel state-dependent dynamic guidance scheme - FeedBack Guidance (FBG)，which uses a state-dependent coefficient to self-regulate guidance amounts based on need. Experiment on the ImageNet benchmark demonstrate its effectiveness.

**Questions:**

- CFG is usually used to control the prompt-following capability of the generated image. Should it measure the text-to-image alignment of the generated image to evaluate the effectiveness of the proposed method?
- What is the meaning of Precision and Recall in the context of the experiment section? What about the $FD_{DinoV2}$?
- The proposed FBG works in a state- and timestep-dependent manner, so how is the extra overhead brought by this scheme compared with the naive CFG scheme?
- The proposed method shows timestep-aware guidance derived from the professional derivation. I'm curious about how the performance comparison is with the simple tiemstep-aware dynamic weighing CFG scheme, like weight scheduler [1], which comes from the empirical observation.
-  The implementation details of the Hybrid CFG-FBG?

[1]  WANG, at, al. Weight-Scheduler: Analysis of classifier-free guidance weight schedulers.

**Ethical Concerns:**

["NO or VERY MINOR ethics concerns only"]

**Final Justification:**

The issue with this work is the inadequacy of the experiment. The author primarily compares the proposed method with the LIG while ignoring other CFG-related methods. In the rebuttal stage, although the author provides some justification for the experiment, it is not fully convincing via the internal experiment without any quantitative results as evidence. In addition, the theoretical derivation of this work is somewhat unclear. Therefore, I have decided to retain my initial rate.

**Limitations:**

yes

**Quality:**

2

**Strengths And Weaknesses:**

**Strengths**:
- Good writing, and easy to follow
- The idea is novel and interesting, and more importantly, supported by solid theoretical proof

**Weaknesses**:
- The experiments are insufficient, without comparison with other CFG-related studies [1,2]
- Some terms (e.g. $FD_{DinoV2}$) have no explanation, and some derivations are skipped too much (e.g., Sec. .3.3)
- The experiments are insufficient, with limited comparative methods.

[1]  WANG, at, al. Weight-Scheduler: Analysis of classifier-free guidance weight schedulers.

[2] Chung, et al. CFG++: Manifold-constrained classifier free guidance for diffusion models.

---

> ### Author Rebuttal · Authors · 2025-07-31
>
> We would like to start by thanking the reviewer for their acknowledgement of the novelty of our approach. Furthermore, we are glad that they found the paper both interesting and easy to follow.
> ## Response to Weakness #1: Comparison to other baselines
> The reason that our main focus was on comparing our method to CFG and LIG is because these two guidance methods are respectively the most popular and the most performant in the literature. We have, internally, played with other alternative fixed schedulers such as those described in the “Analyzing weight-schedulers” paper [1], but our results have always demonstrated that choosing a finite guidance interval as proposed in LIG always surpassed these. Our understanding is that guidance at the beginning of the diffusion process always has a negative impact on FID, which means that only the sine, linear or $\Lambda$ guidance shapes are prominent options. All three are however active too early, having detrimental effect on the performance. Furthermore, we wanted the main focus to be on the spatio-temporal dependence of our guidance scheme, which is why we did not compare to methods such as CFG++, a reference which we have now aded to the related work section.
>
> [1] WANG, at, al. Weight-Scheduler: Analysis of classifier-free guidance weight schedulers.
> ## Response to Weakness #2: Unclear derivations/metrics
> For the description of the metrics, such as FD$_{\text{DinoV2}}$ or Precision/Recall, see our answer to Question #2.
>
> To make sure that the derivations are easily tractable, we have added two additional lines in the derivation of the main body as well as a detailed derivation in the appendix. Equation 7 now reads:
> Seeing that this was pointed out by two of the reviewers, we have added two additional lines in the derivation of the main body as well as a detailed derivation in the appendix. Equation 7 now reads:
> $$
> \\begin{split}
> \\nabla_x\\log p(x_t \\mid c)&=\\nabla_x\\log\\big(p_{\\theta,t}(x_t|c)-(1-\\pi)p_{\theta,t}(x_t)\big)\\\\&=\\frac{1}{p_{\\theta,t}(x_t|c)-(1-\\pi)p_{\\theta,t}(x_t)}\\big(\\nabla_x p_{\\theta,t}(x_t|c)-(1-\\pi)\nabla_x p_{\\theta,t}(x_t)\\big)~\\\\&= \\nabla_x\\log p_{\\theta,t}(x_t) +\\lambda(x_t, t)\\Big(\\nabla_x\\log p_{\\theta,t}(x_t \\mid c)-\\nabla_x\\log p_{\\theta,t} (x_t)\\Big) ~\\
> \\end{split}
> $$
> We have also added an additional inline equation:
> > "To obtain the last equation, we make use of the identity $\nabla_x p(x) = p(x) \nabla_x \log p(x)$ and then reorder the terms to obtain a familiar looking guidance equation."
>
> We thank the reviewer for pointing this out, seeing that the theoretical underpinning of our approach is fundamental to our work it is of uttermost important that the derivation of our guidance scheme is clear.
> ## Response to Weakness #3: Insufficient Experiments
> We wish to address the reviewer's concern that the small-scale experiments may not be sufficient for assessing the method's significance. We agree that strengthening the paper in this regard would significantly improve its quality and impact.
> ### Extension from small EDM2-XS to larger EDM2-L
> To address this, during the rebuttal period we conducted the same analysis previously performed with the EDM2-XS model using the much larger EDM2-L model, providing more comprehensive and qualitative insights. The initial results, computed on 10k generated images, are presented in Table 1 below. The extension towards 50k generated images, as well as the addition of the Precision and Recall metrics, is under way but will not be ready in time for the initial rebuttal.
> We can observe that the results on EDM2-L are more similar for the different methods, compared to the small model. This is because the EDM2-L model is far more qualitative and requires much less guidance, a result expected from our interpretation of guidance as an error correcting scheme. As a consequence, all guidance approaches are more similar to each other.
>
> **Table 1: Performance comparison on EDM2-L (10k images)**
> | Method | FID | FD$_{\text{DinoV2}}$
> |--------|---------|----------------------|
> | CFG | 8.15 | 119.70 |
> | LIG | 7.78 | **112.68** |
> | FBG |**7.75** | 114.90 |
>
> ### Additional T2I results with Stable Diffusion 2 on MS-COCO 30k
> Similarly, to further support our findings in the T2I setting, we evaluated the FID, FD$_{\text{DinoV2}}$ alongside the average CLIP and Aesthetic Scores for different guidance schemes (CFG, LIG, and our proposed FBG) using Stable Diffusion 2 on the standard MS-COCO 30k dataset. These results, currently limited to a subset of 3k prompts, are reported in Table 2. Results on all 30k prompts will become available next week.
> The MS-COCO30k dataset contains prompts that originate from captioned images, which all remain fairly standard. We expect our approach to work even better in context where the prompts are much more varied. To remain consistent with the literature we however still chose to evaluate on this standard benchmark, on which FBG still slightly outperforms alternative approaches.
>
> **Table 2: Quantitative evaluation of T2I on MS-COCO benchmark (3k images)**
> | Method | FID | FD$_{\text{DinoV2}}$ | CLIP-Score| Aesthetic Score|
> |--------|---------|----------------------|-----------|--------|
> | CFG | 19.64 | 53.55 | 0.310 | 5.650 |
> | LIG | 18.81 | 53.55$^*$| 0.311 | 5.742 |
> | FBG |**18.62** | **52.93** | 0.311 |**5.753**|
>
>  $^*$In this setting, the best LIG interval corresponded to applying guidance on the whole trajectory.
>
> We hope that these results help convince the reviewers of the more general validity of the method, as well as of our commitment to incorporate their valid critique to improve the current submission.
> ## Response to Question #1: Prompt alignment metric
> CFG serves not only to improve prompt alignment but also to maintain overall image quality, as DMs underperform on all metrics without guidance. Its key advantage is letting users trade image diversity for stronger alignment by adjusting the guidance scale. This makes FBG’s compatibility with other methods like CFG and LIG particularly valuable: FBG can improve baseline quality, while the guidance scale remains available for users seeking stronger alignment. Given the relevance of metrics like CLIP‑score and Aesthetic scores in T2I, we report these alongside FID and FD in our Stable Diffusion experiments.
> ## Response to Question #2: Metric explanations
> We thank the reviewer for pointing out the absence of explanations regarding the nature of the used metrics. After all, these are not as common as the FID/CLIP-Score and should therefore be explained. To remedy this we have added a sentence describing the FD$_{\text{DinoV2}}$ metric:
> >“The FD-DinoV2, similarly to the FID, measures the Fréchet Distance between generated images and reference images inside the DinoV2 embedding space. The DinoV2 embedding space is trained on a large diverse unlabeled dataset, in stark contrast with the InceptionV3 network used for FID which is solely trained on ImageNet classes and is therefore insensitive to certain features such as human faces. ”
>
> as well as a sentence describing the Precision-Recall metrics:
> > “The precision and recall metrics, very useful to analyze the quality-diversity tradeoff of a method, measure respectively how many of the generated images lie close to the reference image manifold (quality) and inversely how many of the reference image lie close to the generated image manifold (diversity). ”
> ## Response to Question #3: Computational overhead of FBG
> One of the key advantages of the iterative posterior estimation approach is that it comes at negligible additional cost as it solely relies on precomputed quantities, namely the conditional and unconditional score functions already required for the denoising itself. To further emphasize this point we have added a sentence in the main document in section 3.3 mentioning:
> > “A crucial advantage of computing the posterior using the scheme describe above is that it causes negligible computational overhead as all required quantities, in particular $\mathbf{\mu}_c(\mathbf{x}_t|c)$ and $\mathbf{\mu}(\mathbf{x}_t)$, are already computed.”
> ## Response to Question #4: Comparison to other deterministic weight schedulers
> The obtained state- and time-dependent guidance scale shares some aspects with simple predefined guidance schemes, such as LIG or some schemes described in the “Analyzing CFG weight schedulers” work [1]. In particular, we find that our guidance scale is small during the early stages of diffusion, increases during intermediate stages and finally decreases towards later stages. These trends are perfectly in line with the best performing weight-schedulers described in the aforementioned work [1] as well as with the scheduler proposed by LIG.
> ## Response to Question #5: Implementation of FBG_CFG
> The Hybrid FBG-CFG scheme is implemented in the provided code under the name of *‘Hybrid_CFG_FBG’*. We have further added an algorithm, as well as a paragraph in the appendix section describing how precisely the two guidance schemes can be combined. Essentially, the easiest way to understand this hybrid scheme is to see the total guidance scale as the sum of the two terms, i.e. $\lambda_{\text{tot}}(x,t) = \lambda_{\text{FBG}}(x,t) + \lambda_{\text{CFG}}$.
> ## Summary
> We appreciate the reviewer's concerns and questions, and are convinced our responses as well as the additional experimental results have the potential to improve the manuscript considerably. We hope the reviewer reassesses the manuscript in light of this rebuttal, to see whether it warrants increasing their original score.

---

> > ### Comment · Reviewer_maqm · 2025-08-05
> >
> > Thanks for the author's detailed explanation and additional experimental results. However, I decide to keep my initial rating. The primary reason is that the experiment is not adequate. The CFG mechanism in diffusion models is an active research area, and the work should include comparisons with more competing methods. While the authors provided some justification, the internal results are not fully convincing.  Furthermore, just I mentioned in my initial comment, the derivation and writing of this paper also requires further refinement.

---

> > > ### Author Response · Authors · 2025-08-08
> > >
> > > ### Thank you for your reply!
> > >
> > > We sincerely thank the reviewer for acknowledging our efforts in providing detailed explanations and additional experimental results.
> > >
> > > We are, of course, sorry to hear that the additional experiments did not sufficiently address your concerns. We fully agree that, given the rapid progress in guidance for diffusion models, testing against a broader set of competing methods is important. In our initial submission, we did not include weight schedulers such as those in Wang et al. [1], as our focus was on benchmarking against more widely adopted approaches like LIG. By introducing $t_{\text{start}}$ and $t_{\text{end}}$ , LIG can better adapt to scheduler- and setting-dependent regions of diffusion, whereas [1] employs a fixed symmetry around $t=0.5$. We considered this flexibility a key feature when selecting LIG as a baseline for benchmarking our self-regulated feedback guidance mechanism.
> > >
> > > However, we understand your interest in a more direct empirical comparison with the weight schedulers proposed in [1]. To address this, we plan to include additional benchmarks comparing our approach against the best-performing schedulers from [1] (linear and cosine), as well as the CFG++ baseline. Given the short rebuttal timeline, these experiments cannot be completed in time for this round, but they will help provide a more comprehensive evaluation.
> > >
> > > Regarding the writing and derivations, we carefully revised these sections based on feedback from all reviewers and trust that these changes have improved clarity.
> > >
> > > Once again, we thank the reviewer for their thoughtful and constructive comments. We remain confident that the proposed method is simple, practical, and broadly applicable, with consistent performance across diverse settings. We believe these qualities make it a valuable contribution that is ready for acceptance, and we look forward to building on it in future work.
> > >
> > > [1] WANG, at, al. Weight-Scheduler: Analysis of classifier-free guidance weight schedulers.

---

### Official Review · Reviewer_o9ey · 2025-06-30

**Clarity:** 3
**Significance:** 2
**Originality:** 3
**Rating:** 3
**Confidence:** 4

**Summary:**

This paper introduces a diffusion guidance approach that adaptively adjusts the guidance strength $\lambda$ during sampling. The core idea lies in assuming an additive factorization of the learned conditional into true conditional and unconditional distributions. The guidance scale can then be derived by letting the model to estimate the posterior $\log p(c|\mathbf{x}_t)$. The method has been evaluated on class-conditioned and text-conditioned image generation, demonstrating enhanced performance in terms of quality-diversity tradeoff compared with CFG and LIG.

**Questions:**

Q1. It is still not clear that empirically how to select $t_0$ and $t_1$. More specifically, $t_0$ and $t_1$ should also be dynamically adjusted for different prompts instead of simply tuned as hyperparameters which potentially leads to suboptimal performance.

Q2. How does the proposed approach perform on diffusion models with larger scales? Can it be adopted for flow matching models?

Q3. Can the authors provide more comprehensive comparisons on text-to-image task? This is a critical venue to showcase the advantage of the proposed approach, given that CFG is commonly adopted as an effective and easy-to-use guidance approach.

Q4. It is unclear how $\lambda$ in Eq. 8 is derived. Could the authors elaborate more on the details?

Q5. I am still confused on the rationale of the additive factorization of $p_{\theta,t}(x_t|c)$. It would be helpful if the authors can elaborate more on the comparison of additive versus multiplicative. In particular the statement in line 156 "gives non-zero probability to regions outside the conditional $c$" seems confusing.

**Ethical Concerns:**

["NO or VERY MINOR ethics concerns only"]

**Final Justification:**

The rebuttal partially addresses my questions. However these concerns remain.

The experiments on larger-scale model are not completed. For the numbers provided, the performance is quite close for all guidance approaches, which raises uncertainty on the actual scalability and applicability of the proposed approach.

As a guidance method, I believe both effectiveness in terms of performance and convenience in terms of to what degree the method is easy to adopt are crucial factors that affect its applicability. In this respect, the additionally introduced hyperparameters further pose challenges to the practitioners who want to choose a guidance approach to perform for their downstream application. It would be more helpful if more in-depth analyses such as how these hyperparameters affect different tasks.

There are also other minor issues about presentation that are possibly fixable but indeed have raised concerns during reading. e.g., incorporating more details in the derivation to avoid confusion.

Therefore I give a borderline score and believe that the paper can be carefully revised and further improved.

**Limitations:**

yes

**Quality:**

2

**Strengths And Weaknesses:**

## Strengths

1. The paper is well motivated by arguing that the open-loop guidance methods like CFG are suboptimal. At the same time, the fixed guidance scale is CFG does not take into account the dynamic nature throughout the diffusion sampling.

2. The idea of dynamically adjusting the guidance weight based on the model's self-estimated confidence of the class posterior seems interesting.

3. The experiments demonstrate interesting observations of the dynamic nature of the proposed approach in terms of enforcing different guidance scale at different diffusion steps and with different difficulties of the prompt.

## Weaknesses

1. The experimental evaluation is more or less illustrative and lacks comprehensiveness. In particular, the class conditional generation experiments are only performed using EDM2-XS, while the text-conditioned generation experiments have no numerical results. It is thus unclear how much gain can be achieved by the proposed approach.

2. While being theoretically motivated, heuristics have been adopted such as introducing additional hyperparameters like $\delta$ and $\pi$ which leads to difficulty in hyperparameter tuning. By constrast, baselines such as CFG does not require extensive tuning of these additional hyperparameters, which seems like an overhead. More discussions should be conducted to show that introducing these variables leads to gains that outweigh the inconvenience.

3. The presentation leads to various vague and confusing statements. Please refer to Q4-Q5 in Questions.

Minor: multiple typos, e.g., Algorithm 1, line 1, the blue highlight should be $\mu_\theta(x_t)$. Table 1, what is DCFG? Line 167, should be with $\lambda(x_t,t)$ as guidance scale.

---

> ### Author Rebuttal · Authors · 2025-07-31
>
> First and foremost, we would like to thank the reviewer for their in depth review of our manuscript. We appreciate that the reviewer recognizes the novelty of the dynamic posterior dependent guidance scale. We are happy that they found the work interesting, and also thank them for having taken the time to come up with interesting and challenging questions that, in our eyes, will improve the paper.
> ## Response to Weakness #1: Requirement for more exhaustive experiments
> We first address the reviewer's concern that the small-scale experiments may not be sufficient for assessing the method's significance. We agree that strengthening the paper in this regard would significantly improve its quality and impact.
> ### Extension from small EDM2-XS to larger EDM2-L
> To address this, during the rebuttal period we conducted the same analysis previously performed with the EDM2-XS model using the much larger EDM2-L model, providing more comprehensive and qualitative insights. The initial results, computed on 10k generated images, are presented in Table 1 below. The extension towards 50k generated images, as well as the addition of the Precision and Recall metrics, is under way but will not be ready in time for the initial rebuttal.
> We can observe that the results on EDM2-L are more similar for the different methods, compared to the small model. This is because the EDM2-L model is far more qualitative and requires much less guidance, a result expected from our interpretation of guidance as an error correcting scheme. As a consequence, all guidance approaches are more similar to each other.
>
> **Table 1: Performance comparison on EDM2-L (10k images)**
> | Method | FID | FD$_{\text{DinoV2}}$
> |--------|---------|----------------------|
> | CFG | 8.15 | 119.70 |
> | LIG | 7.78 | **112.68** |
> | FBG |**7.75** | 114.90 |
>
> ### Additional T2I results with Stable Diffusion 2 on MS-COCO 30k
> Similarly, to further support our findings in the T2I setting, we evaluated the FID, FD$_{\text{DinoV2}}$ alongside the average CLIP and Aesthetic Scores for different guidance schemes (CFG, LIG, and our proposed FBG) using Stable Diffusion 2 on the standard MS-COCO 30k dataset. These results, currently limited to a subset of 3k prompts, are reported in Table 2. Results on all 30k prompts will become available next week.
> The MS-COCO30k dataset contains prompts that originate from captioned images, which all remain fairly standard. We expect our approach to work even better in context where the prompts are much more varied. To remain consistent with the literature we however still chose to evaluate on this standard benchmark, on which FBG still slightly outperforms alternative approaches.
>
> **Table 2: Quantitative evaluation of T2I on MS-COCO benchmark (3k images)**
> | Method | FID | FD$_{\text{DinoV2}}$ | CLIP-Score| Aesthetic Score|
> |--------|---------|----------------------|-----------|--------|
> | CFG | 19.64 | 53.55 | 0.310 | 5.650 |
> | LIG | 18.81 | 53.55$^*$| 0.311 | 5.742 |
> | FBG |**18.62** | **52.93** | 0.311 |**5.753**|
>
> $^*$In this setting, the best LIG interval corresponded to applying guidance on the whole trajectory.
>
> We hope that these results help convince the reviewers of the more general validity of the method, as well as of our commitment to incorporate their valid critique to improve the current submission.
> ## Response to Weakness #2: Introduction of $t_0$ and $t_1$
> Our Feedback Guidance (FBG) approach, like Limited Interval Guidance (LIG), has three hyperparameters, making initial tuning harder than CFG. However, unlike LIG, FBG is derived from first principles, offering insights into why guidance is needed. Its main advantage is that it automatically detects when and where strong guidance is required, removing the need for a changeable guidance scale—unlike CFG or LIG, which use fixed schedulers independent of prompt or state. Once valid hyperparameters are set at the model/application level, FBG self‑regulates guidance across prompts and timesteps, focusing on parts needing the strongest correction, as shown in our T2I results. We acknowledge the concern about introducing these hyperparameters, which is why we added:
> >“The hyperparameters introduced for FBG are conceptually interpretable and need to be set only once. They allow the model to self-regulate guidance strength across prompts and timesteps, reducing the need for extensive prompt specific tuning compared to fixed-scheduler approaches.”
> ## Response to Weakness #3: Unclear statements
> We have gone through the manuscript in close detail again to remove any possibly confusing statements, such as the typos mentioned (such as the “DCFG” present in Table 1, which was the previous acronym we had internally used for our FBG scheme and which was unfortunately left unchanged before the submission). In this regard, we also spend close attention to Q4 and Q5, for which additional sentences/derivations have been added to the main document.
> ## Response to Question #1: Selection of $t_0$ and $t_1$
> While $t_0$ and $t_1$ require manual tuning, we designed them to be highly interpretable, simplifying this process. As outlined, they correspond to the average moments guidance starts and stops (similar to $t_{\text{start}}$ and $t_{\text{end}}$ in LIG), and intuitive reasoning—e.g., earlier or wider intervals producing greater changes—helps calibrate them. Figures 3a and 8a further show that across a reasonable range of $t_0$/$t_1$ values, FBG outperforms CFG on EDM2‑XS using FD$_{\text{DinoV2}}$ and FID. While making $t_0$ and $t_1$ dynamic (e.g., automatically adjust to earlier timesteps for low‑frequency features like landscapes) is a promising direction we are highly interested in, we believe our fixed‑parameter approach already represents a significant advance over static guidance schemes.
> ## Response to Question #2: Larger scale models + Flow Matching models
> For the analysis of larger scale models we refer to our answer of Weakness #1.
> Regarding the adaptability to Flow Matching Models, although left as future work, it is our belief that a very similar Feedback Guidance scheme can easily be derived and described in this context. The novel ideas introduced, such as the dynamic posterior dependent guidance scale, are in no way connected to the specific choice of diffusion models described in this paper. We are very excited about future works looking into extending the framework of Feedback Guidance to Flow Matching models, Stochastic Interpolators or Schrödinger Bridges.
> ## Response to Question #3: Evaluation of T2I
> In the context of T2I, we have focused on obtaining FID/FD$_{\text{DinoV2}}$ values as well as CLIP and Aesthetic Scores on MS-COCO 30k, on which, similarly to what was obtained using EDM2-XS, FBG outperforms CFG and performs on par with LIG. We also agree with the reviewer that careful visualizations are key to understanding a paper. To improve on the current, limited, illustrative examples, we have have prepared multiple additional images similar to Fig. 4a and Fig. 5 that will be added to the appendices. Unfortunately, to remain in line with the NeurIPS rebuttal guidelines, we can not share them at this stage.
> ## Response to Question #4: Unclear derivation of the dynamic guidance scale
> Seeing that this was pointed out by two of the reviewers, we have added two additional lines in the derivation of the main body as well as a detailed derivation in the appendix. Equation 7 now reads:
> $$
> \\begin{split}
> \\nabla_x\\log p(x_t \\mid c)&=\\nabla_x\\log\\big(p_{\\theta,t}(x_t|c)-(1-\\pi)p_{\theta,t}(x_t)\big)\\\\&=\\frac{1}{p_{\\theta,t}(x_t|c)-(1-\\pi)p_{\\theta,t}(x_t)}\\big(\\nabla_x p_{\\theta,t}(x_t|c)-(1-\\pi)\nabla_x p_{\\theta,t}(x_t)\\big)~\\\\&= \\nabla_x\\log p_{\\theta,t}(x_t) +\\lambda(x_t, t)\\Big(\\nabla_x\\log p_{\\theta,t}(x_t \\mid c)-\\nabla_x\\log p_{\\theta,t} (x_t)\\Big) ~\\
> \\end{split}
> $$
> We have also added an additional inline equation:
> >  "To obtain the last equation, we make use of the identity $\nabla_x p(x) = p(x) \nabla_x \log p(x)$ and then reorder the terms to obtain a familiar looking guidance equation."
>
> We thank the reviewer for highlighting this, as the theoretical underpinning is fundamental to our work, making a clear derivation of our guidance scheme is essential.
> ## Response to Question #5: Rationale behind additive vs. multiplicative assumption
> The interpretation of guidance schemes through error assumption models is in our eyes one of the most relevant contributions of our work. We agree with the reviewer that explaining the rationale behind the choice of an additive error model is important, which is why we have added two sentences in the main body.
> >“The additive assumption can be seen as less restrictive than the multiplicative one as it allows the learned conditional distribution $p_{\theta,t}(x_t|c)$ to be non-zero in regions where the true conditional distribution $p_t(x_t|c)$ is zero, a feat the multiplicative assumption is incapable of. Due to the joint training pipeline, and the fact that training pairs often contain more than a single element, such an overlap of the learned distributions is in practice highly likely.“
>
> We would however like to emphasize that our point is not that the additive error assumption is fully representative of the true error of the learned model, but merely that this new assumption might lead to different properties for the guidance scheme. In the future, analyzing different, more intricate, error models might lead to further findings, which we look forward to analyzing.
>
> ## Summary
> We appreciate the reviewer's concerns and questions, and are convinced our responses as well as the additional experimental results have the potential to improve the manuscript considerably. We hope the reviewer reassesses the manuscript in light of this rebuttal, to see whether it warrants increasing their low original score.

---

> > ### Comment · Reviewer_o9ey · 2025-08-05
> >
> > Thank you for the response. I tend to raise the score to 3. However, this paper still seems borderline to me. Here are the primary reasons.
> >
> > The experiments on larger-scale model are not completed. For the numbers provided, the performance is quite close for all guidance approaches, which raises uncertainty on the actual scalability and applicability of the proposed approach.
> >
> > As a guidance method, I believe both effectiveness in terms of performance and convenience in terms of to what degree the method is easy to adopt are crucial factors that affect its applicability. In this respect, the additionally introduced hyperparameters further pose challenges to the practitioners who want to choose a guidance approach to perform for their downstream application. It would be more helpful if more in-depth analyses such as how these hyperparameters affect different tasks.
> >
> > There are also other minor issues about presentation that are possibly fixable but indeed have raised concerns during reading. e.g., incorporating more details in the derivation to avoid confusion.

---

> > > ### Author Response · Authors · 2025-08-05
> > > **Thank you for your reply**
> > >
> > > ## Thank you for your reply!
> > > We sincerely thank you for your response and for raising your score to 3.
> > >
> > > We fully agree that the points you raised, such as assessing our method's applicability across more diverse downstream tasks, and conducting a deeper analysis on different choices of hyperparameters are highly relevant. We view these as important directions for future research and believe that exploring them will further strengthen the practical value of our method.
> > >
> > > We also appreciate your feedback on the presentation and agree that ensuring the derivations and explanations are as clear as possible is of utmost importance.
> > >
> > > Thank you again for your thoughtful comments and for maintaining a critical perspective, which we believe will ultimately help improve the paper.

---

### Official Review · Reviewer_qaVs · 2025-07-02

**Clarity:** 3
**Significance:** 3
**Originality:** 3
**Rating:** 5
**Confidence:** 4

**Summary:**

This paper addresses the limitation of classifier-free guidance (CFG) where its use of constant guidance scale throughout the process can reduce generation diversity. This paper proposes a novel feedback guidance (FBG) method that can adaptably determine the guidance scale throughout the denoising process, based on the feedback of a quality estimation of its current predictions. This makes FBG state-dependent (e.g., closer to conditional guidance leads to lower scale, while closer to unconditional guidance leads to higher scale; higher scale for complex prompts, while lower for easier prompts). FBG is also time-dependent, where it is only active in the middle of the denoising process, due to the limitation of injecting CFG in the early and late stages. Experiments are conducted on ImageNet using EDM2-XS model, and results have demonstrate superior performance over CFG and comparable performance with LIG.

**Questions:**

Please see the previous strengths and weaknesses section. Addressing the four aspects mentioned in the weaknesses section can effectively address my concerns.

**Ethical Concerns:**

["NO or VERY MINOR ethics concerns only"]

**Final Justification:**

The rebuttal has effectively addressed my concerns by providing additional clarifications and results; combined with the advantages listed in the strengths section, I would maintain my support of this paper's acceptance.

**Limitations:**

yes

**Quality:**

3

**Strengths And Weaknesses:**

**Strengths**:

1. The task is well-motivated. Considering the important role guidance plays in diffusion models, improving on the standard CFG can provide wide benefits to the field.
2. The presentation is overall clear, apart from some potential small improvements. This makes the paper easy to follow. Especially, the visualizations are helpful for delivering the insights and demonstrating the rationale.
3. The proposed method is novel and effective, surpassing CFG, while achieving comparable results with LIG.
4. The proposed method is also highly adaptable, resulting in using state- and time-dependent guidance scales. Another practical benefit is its combinable with other guidance schemes.
5. Finally, a notable advantage of this work is its solid theoretical underpinnings.

**Weaknesses**:

1. Due to the design of the proposed FBG, it necessitates the computation of the “feedback” to determine the guidance scale. This will bring additional computational costs in addition to the original inference process.
2. It’d better of Figure 1 can include labels or annotations of each line in the plot or as more detailed descriptions in the caption to improve on its clarity. Similarly, in Table 1, it is suggested to explain in the caption what underline and bold mean when they are used for highlighting the numbers.
3. FBG’s performance does not always surpass LIG. However, this is not a major concern as the performance is comparable and FBG benefits from its solid theoretical backgrounds.
4. Would the proposed method work if using DiT as the diffusion backbone? It would be interesting to see more experimental results using a variety of settings such as whether the proposed FBG is effective in larger architectures, in one-step generative models, etc. The inclusion of these and the proof of its scalability would make this work more influential.

---

> ### Author Rebuttal · Authors · 2025-07-31
>
> We would first like the reviewer for the in depth reading of our manuscript. In particular, we are glad that they appreciate the novelty of our novel state- and time-dependent guidance approach as well as its solid theoretical underpinnings.
>
> ## Response to Weakness #1: Computational overhead
>
> One of the key advantages of the iterative posterior estimation approach is that it comes at negligible additional cost as it solely relies on precomputed quantities, namely the conditional and unconditional score functions already required for the denoising itself. To further emphasize this point we have added a sentence in the main document in section 3.3 mentioning:
> > *“A crucial advantage of computing the posterior using the scheme describe above is that it causes negligible computational overhead as all required quantities, in particular $\mathbf{\mu}_c(\mathbf{x}_t|c)$ and $\mathbf{\mu}(\mathbf{x}_t)$, are already computed.”*
>
> ## Response to Weakness #2: Description of Fig. 1/Table 1
>
> We thank the reviewer for this valuable advice. We have included descriptive labels for the three lines present in Fig 1. which describe the three trajectories using the adjectives: “good”, “okay” and “poor”. Their meaning in the context of the generative trajectories is then further explained in the caption:
> >*"Figure1: Illustrative diffusion trajectories and their hypothetical guidance scales in a 1D setting. Trajectories located further from the mode close to the decision window, such as the trajectories denoted by "okay" and "poor", receive higher guidance. On the contrary trajectories that are clearly going to the right mode, such as that denoted by "good", receive negligible guidance"*
>
>  Regarding Table 1, the caption has been modified to explain the use of bold and underlined:
>  > *"We underline the best performing method per sampler, and further emphasize the best overall method using bold values."*
>
> ## Response to Weakness #3: FBG compared to LIG
>
> We acknowledge that when measured using the FID, FBG does not outperform LIG, as mentioned in the abstract and recognized by the reviewer. Indeed, we solely claim performance comparable to LIG with as main advantage a robust theoretical underpinning.
>
> ## Response to Weakness #4: Testing FBG on larger architectures
> Finally we wish to address the reviewer's concern that the small-scale experiments may not be sufficient for assessing the method's significance. We agree that strengthening the paper in this regard would significantly improve its quality and impact. It should also be noted that the described theoretical framework is certainly not limited to a specific architecture and that FBG is fully compatible with models such as DiT. We believe this versatility, gained by rigorous mathematical foundations, to be one of the main strengths of our paper.
>
> ### Extension from small EDM2-XS to larger EDM2-L
> To address this, during the rebuttal period we conducted the same analysis previously performed with the EDM2-XS model using the much larger EDM2-L model, providing more comprehensive and qualitative insights. The initial results, computed on 10k generated images, are presented in Table 1 below. The extension towards 50k generated images, as well as the addition of the Precision and Recall metrics, is under way but will not be ready in time for the initial rebuttal.
> We can observe that the results on EDM2-L are more similar for the different methods, compared to the small model. This is because the EDM2-L model is far more qualitative and requires much less guidance, a result expected from our interpretation of guidance as an error correcting scheme. As a consequence, all guidance approaches are more similar to each other.
>
> **Table 1: Performance comparison on EDM2-L (10k images)**
>
> | Method | FID | FD$_{\text{DinoV2}}$
> |--------|---------|----------------------|
> | CFG | 8.15 | 119.70 |
> | LIG | 7.78 | **112.68** |
> | FBG |**7.75** | 114.90 |
>
> ### Additional T2I results with Stable Diffusion 2 on MS-COCO 30k
> Similarly, to further support our findings in the T2I setting, we evaluated the FID, FD$_{\text{DinoV2}}$ alongside the average CLIP and Aesthetic Scores for different guidance schemes (CFG, LIG, and our proposed FBG) using Stable Diffusion 2 on the standard MS-COCO 30k dataset. These results, currently limited to a subset of 3k prompts, are reported in Table 2. Results on all 30k prompts will become available next week.
> The MS-COCO30k dataset contains prompts that originate from captioned images, which all remain fairly standard. We expect our approach to work even better in context where the prompts are much more varied. To remain consistent with the literature we however still chose to evaluate on this standard benchmark, on which FBG still slightly outperforms alternative approaches.
>
> **Table 2: Quantitative evaluation of T2I on MS-COCO benchmark (3k images)**
> | Method | FID | FD$_{\text{DinoV2}}$ | CLIP-Score| Aesthetic Score|
> |--------|---------|----------------------|-----------|--------|
> | CFG | 19.64 | 53.55 | 0.310 | 5.650 |
> | LIG | 18.81 | 53.55$^*$| 0.311 | 5.742 |
> | FBG |**18.62** | **52.93** | 0.311 |**5.753**|
>
> $^*$In this setting, the best LIG interval corresponded to applying guidance on the whole trajectory.
>
> We hope that these results help convince the reviewers of the more general validity of the method, as well as of our commitment to incorporate their valid critique to improve the current submission.
>
> ## Summary
> We appreciate the reviewer's concerns and questions, and are convinced our responses as well as the additional experimental results have the potential to improve the manuscript considerably. We thank the reviewer for reassessing the manuscript in light of this rebuttal.

---

> > ### Comment · Reviewer_qaVs · 2025-08-05
> >
> > Thanks for the rebuttal with additional explanation and results. They have effectively addressed my concerns; combined with the advantages listed in the strengths section, I would maintain my support of this paper's acceptance.

---

### Official Review · Reviewer_UFxC · 2025-07-03

**Clarity:** 3
**Significance:** 3
**Originality:** 3
**Rating:** 4
**Confidence:** 4

**Summary:**

This paper proposes a state-dependent guidance scale method, which can automatically adjust the current guidance scale according to the posterior distribution after sampling to obtain the optimal value. Experiments on class-to-image (ImageNet 512x512) and text-to-video tasks have demonstrated that this method achieves significantly better results than the conventional CFG (Classifier-Free Guidance).

**Questions:**

1. Please add more experiment on text-to-video generation, DiT-XL/2 on ImageNet 256x256, as the proposed method is training-free and data-free, conducting on video model is accepetable.
2. Impact of Prompt. In text-to-video, what is the result if prompt is missing, wrongly provided, or more detailed provided.
3. More Analysis of more iterations of mix scores + take step + guidance scale.
4. More experiments of few-step generation when the number of total denoising steps are decreased.
5. Add more evaluation metrics such as HPS v2, aesthetic score, clip score.

**Ethical Concerns:**

["NO or VERY MINOR ethics concerns only"]

**Final Justification:**

The authors initially conducted experiments only on small models. The rebuttal results on large models such as EDM2-L demonstrate that all guidance approaches are more similar to each other.​
As a training-free general CFG method, it is feasible to conduct quick evaluations on video models to validate its generalization. However, the authors failed to address this issue.​
Considering the lack of results on HPSv2, limited improvements in metrics such as FID and CLIP score, along with the introduced complex pipeline (mix scores + take step + update guidance), I can only retain my score and judge it to be between 3 and 4.

**Limitations:**

yes

**Quality:**

3

**Strengths And Weaknesses:**

Strengths​
1. Quality: The training-free, data-free, and online method boosts generation quality based on CFG with simple, reproducible code. Extra forward pass costs are acceptable for quality gains.​ The experiment includes both FID and FID-dino on sampled about 10k images, which is reliable.
2. Clarity: Methodology and implementation are clearly presented.​
3. Significance: Offers an efficient way to improve generative model performance without extra training/data.​
4. Originality: Automatically adjusting guidance scale via posterior distribution is a novel approach.​

Weaknesses​
1. Experiments only on EDM-XS due to limited computing power fail to match the goal of achieving optimal CFG, raising doubts about the method's necessity for accurate models.​ Small-scale experiments undermine the method's significance; testing on DiT-XL/2, SiT is required.​
2. Only evaluating with FID and FID-dino is not enough. If the proposed guidance scale method could work, it should also improve human preference.

---

> ### Author Rebuttal · Authors · 2025-07-31
>
> We would first like to thank the reviewer for their detailed analysis, which recognizes the quality, originality and significance of our work. In particular, we appreciate the reviewer’s acknowledgments of our efforts to provide clear and usable code, as well as realizing that our proposed scheme comes with negligible computational overhead when compared to CFG.
> ## Response to Weakness #1: limited experiments
> We first address the reviewer's concern that the small-scale experiments may not be sufficient for assessing the method's significance. We agree that strengthening the paper in this regard would significantly improve its quality and impact.
> ### Extension from small EDM2-XS to larger EDM2-L
> To address this, during the rebuttal period we conducted the same analysis previously performed with the EDM2-XS model using the much larger EDM2-L model, providing more comprehensive and qualitative insights. The initial results, computed on 10k generated images, are presented in Table 1 below. The extension towards 50k generated images, as well as the addition of the Precision and Recall metrics, is under way but will not be ready in time for the initial rebuttal.
> We can observe that the results on EDM2-L are more similar for the different methods, compared to the small model. This is because the EDM2-L model is far more qualitative and requires much less guidance, a result expected from our interpretation of guidance as an error correcting scheme. As a consequence, all guidance approaches are more similar to each other.
>
> **Table 1: Performance comparison on EDM2-L (10k images)**
>
> | Method | FID | FD$_{\text{DinoV2}}$
> |--------|---------|----------------------|
> | CFG | 8.15 | 119.70 |
> | LIG | 7.78 | **112.68** |
> | FBG |**7.75** | 114.90 |
>
> ### Additional T2I results with Stable Diffusion 2 on MS-COCO 30k
> Similarly, to further support our findings in the T2I setting, we evaluated the FID, FD$_{\text{DinoV2}}$ alongside the average CLIP and Aesthetic Scores for different guidance schemes (CFG, LIG, and our proposed FBG) using Stable Diffusion 2 on the standard MS-COCO 30k dataset. These results, currently limited to a subset of 3k prompts, are reported in Table 2. Results on all 30k prompts will become available next week.
> The MS-COCO30k dataset contains prompts that originate from captioned images, which all remain fairly standard. We expect our approach to work even better in context where the prompts are much more varied. To remain consistent with the literature we however still chose to evaluate on this standard benchmark, on which FBG still slightly outperforms alternative approaches.
>
> **Table 2: Quantitative evaluation of T2I on MS-COCO benchmark (3k images)**
>
> | Method | FID | FD$_{\text{DinoV2}}$ | CLIP-Score| Aesthetic Score|
> |--------|---------|----------------------|-----------|--------|
> | CFG | 19.64 | 53.55 | 0.310 | 5.650 |
> | LIG | 18.81 | 53.55$^*$| 0.311 | 5.742 |
> | FBG |**18.62** | **52.93** | 0.311 |**5.753**|
>
> $^*$In this setting, the best LIG interval corresponded to applying guidance on the whole trajectory.
>
> We hope that these results help convince the reviewers of the more general validity of the method, as well as of our commitment to incorporate their valid critique to improve the current submission.
> ## Response to Weakness #2: no human evaluation
> We fully agree with the reviewer that in terms of quality metrics, the end goal remains human evaluation. Unfortunately, in the short amount of time available during the rebuttal, performing such an analysis remained unfeasible. Instead, we rely on common practice in the field, which has demonstrated the strong correlation between human preferences and FD metrics, more particularly the FD$_{\text{DinoV2}}$ [1]. In the context of T2I we believe that the a much wider ranger of metrics is preferable, which is why we chose to also provide both CLIP-Score and Aesthetic Scores. We hope that these additional metrics help convince the reviewer of the validity of the proposed evaluation approach.
>
> [1] Stein et al., “Exposing flaws of generative model evaluation metrics and their unfair treatment of diffusion models”
> ## Response to Question 1: experiments on text-to-video?
> Regarding the analysis of larger models, we refer the reviewer to Response to Weakness #1. Unfortunately, due to the short time available during the rebuttal phase, testing in the context of T2V, or on DiT, was not feasible with the computational resources available. We hope that our evaluation setting of choice (in line with the tradition of image benchmarks and metrics, to validate research on diffusion models) remains acceptable to the reviewer.
> ## Response to Question 2: on underspecified prompts
> As no experiments have been performed in the context of T2V, we can not comment on this specific setting. However, the questions remains interesting and insightful in the context of T2I. Our dynamic guidance method adjusts according to the guidance needed for each prompt. This does not mean that less detailed (“easier”) prompts are less well followed; on the contrary, their lack of specificity allows for more diverse secondary features. The key point is that not all prompts require the same level of guidance, and low guidance does not necessarily imply low quality. In this regard FBG provides a key edge over established methods such as CFG which always applies the same amount of guidance regardless of the amount of detail and/or complexity of the prompt. This is well highlighted by the results shown in Fig. 4 of the paper.
>
> ## Response to Questions 3-4: on iterations of: mix scores + take step + update guidance
>
> An important note is that the graph displayed in Fig. 2 is unrolled in time, implying that applying more iterations corresponds to a change in the total number of steps used in the diffusion process. Theoretically, the more steps are used the more accurate the posterior approximation becomes. To avoid any future confusion for readers we have decided to add the following sentence after line 204:
> >“This control diagram is progressively unrolled during the denoising process, implying a succession of computing the score functions, mixing them, applying a denoising step, updating the value of the guidance scale and repeating a fixed amount of times until a fully denoised image is obtained.”
>
> As our main focus lied on a theoretically grounded method, we preferred testing the use of different solvers (2nd order Heun and stochastic) which operate in fundamentally distinct ways. The analysis of varying the number of sampling steps, such as analyzing the case of few-step generation which we considered as non-central to our work, was left as future work.
>
> ## Response to Question 5: add prompt alignment metrics
>
> To beter evaluate the quality of the proposed approach we evaluated the T2I results using a wider range of metrics, including the FID,FD$\_{\text{DinoV2}}$ , CLIP-Score and the Aesthetic Score. We strongly agree with the reviewer that in the case of T2I with the absence of human evaluation, using at least a variety of metrics is key. However, we would like to point out that in the context of Imagenet, which contains almost exclusively well-centered, well-lit images, these metrics can be seen as slightly less relevant than the provided FD$_{\text{DinoV2}}$ and the Precision and Recall curves, which also measure the quality-diversity trade-off of the method [1]. We hope that the reviewer is satisfied by the additional metrics, on which FBG also outperforms concurrent approaches.
>
> ## Summary
> We appreciate the reviewer's concerns and questions, and are convinced our responses as well as the additional experimental results have the potential to improve the manuscript considerably. We hope the reviewer reassesses the manuscript in light of this rebuttal, to see whether it warrants increasing their original score.

---

> > ### Comment · Reviewer_UFxC · 2025-08-07
> >
> > The authors initially conducted experiments only on small models. The rebuttal results on large models such as EDM2-L demonstrate that all guidance approaches are more similar to each other.​
> > As a training-free general CFG method, it is feasible to conduct quick evaluations on video models to validate its generalization. However, the authors failed to address this issue.​
> > Considering the lack of results on HPSv2, limited improvements in metrics such as FID and CLIP score, along with the introduced complex pipeline (mix scores + take step + update guidance), I can only retain my score and judge it to be between 3 and 4.

---

> > > ### Author Response · Authors · 2025-08-08
> > >
> > > ### Thank you for your reply
> > >
> > > We thank the reviewer for their thoughtful feedback and for taking the time to consider our rebuttal.
> > >
> > > Regarding the suggestion to evaluate on text-to-video models, such experiments are unfortunately not feasible for us at this stage. Nevertheless, we believe that the results on EDM2-L, while showing smaller margins, still indicate that FBG consistently outperforms concurrent approaches. Moreover, guidance is often applied in settings where, as in the EDM2-XS case, a strong conditional model is not available (e.g., text-to-image generation on diverse, user-chosen prompts), where the advantages of our approach are more pronounced.
> > >
> > > Concerning the complexity of the pipeline, the only additional component is the *update guidance* step, which comes at negligible computational cost; *mix scores* and *take step* are already part of any denoising scheme. In fact, a key strength of our method is its simplicity: it uses only the model itself to decide when guidance is beneficial for a specific trajectory.
> > >
> > > While we understand your decision to maintain your score, we had hoped that our rebuttal (presenting evaluations on larger models containing aesthetic metrics, results on standard text-to-image benchmarks, and clarifying that our scheme incurs no additional cost) would address your concerns. We remain confident that the contribution is both practical and well-motivated, and we hope this reply helps convey why we believe it merits acceptance.

---

### Note · Authors · 2025-08-11

We would like to sincerely thank all reviewers for their constructive and insightful feedback. Their comments have helped us improve and clarify our presentation as well as broaden our experimental evaluation, ultimately strengthening both the technical content and the clarity of the paper.

During the rebuttal process, we have run additional quantitative analyses of T2I generation on MS-COCO, extending our evaluation beyond FID and FD$_{\text{DinoV2}}$ to include Aesthetic and CLIP scores, and we have added experiments on larger models such as EDM2-L. These results are still preliminary, as the short rebuttal timeline did not allow for a full-scale analysis with FID values at 50k images, but they already indicate promising performance of our scheme. We have also provided clarifications where needed, such as highlighting that our Feedback Guidance scheme incurs negligible computational overhead compared to widespread methods such as CFG. Furthermore, we have expanded the theoretical derivations, which we consider to be one of the key strengths of our work relative to concurrent literature.

Overall, we believe that the novelty of introducing a state- and time-dependent guidance scale, together with its strong theoretical foundations, makes this work a valuable contribution to the field. Its demonstrated performance on the considered benchmarks further supports its relevance, and we hope the revisions made during the rebuttal process help to clearly convey its potential impact.

---

### Decision · Program_Chairs · 2025-09-17

**Decision:**

Accept (poster)

**Comment:**

This paper proposes a guidance approach for diffusion models called Feedback Guidance (FBG). In contrast to classifier-free guidance (CFG), which treats the guidance strength as a fixed hyperparameter, FBG dynamically adapts the guidance strength over the denoising trajectory. The paper provides a theoretical framework and derivation of the approach.

Empirically, the authors evaluate FBG on class-conditional generation on ImageNet 512x512 using the EDM2-XS model. They compare FBG to vanilla classifier-free guidance and limited interval guidance (LIG), which is another approach to dynamic guidance that serves as a strong baseline. Overall, FBG outperforms CFG and performs similarly to LIG. FBG has better theoretical grounding than LIG, which is a heuristic approach.

The paper includes interesting results showing how the guidance scale changes during denoising for different prompts, which demonstrates that it can differ on a per-prompt basis.

Reviewer UFxC mentioned that the paper was clear, novel, and seems useful as it does not require extra training. This reviewer raised concerns regarding the scale of the experiments (only performed on EDM2-XS), the lack of evaluation metrics such as HPSv2, CLIPScore, and Aesthetic score, and the lack of text-to-video results. The authors provided some experiments on the larger EDM2-L model in their rebuttal, and added results using FBG for Stable Diffusion 2 on MS-COCO30k. In general, I do not think that text-to-video experiments are necessary for such a paper, as this is not commonly done in related work.

Reviewer qaVs noted that the paper is novel, well-motivated with solid theoretical underpinnings, and clearly written. The reviewer also noted that the proposed method is novel and effective, outperforming CFG and attaining similar performance to LIG.

They raised concerns about the computational cost of FBG and the lack of evaluation of a DiT diffusion backbone, a larger model, or a one-step generative model. The rebuttal addressed this reviewer’s concerns.

Reviewer o9ey noted that the paper is well-motivated and the idea of dynamically adjusting the guidance weight is interesting.

This reviewer raised concerns regarding the small model EDM2-XS used in the evaluation, the hyperparameters that need to be tuned for the FBG, and the clarity of parts of the exposition. In the rebuttal, the authors provided results on EDM2-L and Stable Diffusion 2. While the results show that different guidance methods perform somewhat similarly to each other, FBG still behaves similarly to LIG and both perform better than CFG. The reviewer was not fully convinced by the larger-scale results, but they nevertheless show that FBG does work in this setting. In the rebuttal, the authors acknowledged that their method has three hyperparameters, but LIG does too, so this does not seem like a key issue.

Reviewer maqm noted that the paper is well-written, novel, and backed by theory. This reviewer raised concerns about insufficient comparisons with other CFG methods (Weight-Scheduler and CFG++) and missing explanations in the derivations.

I agree that adding quantitative comparisons to more guidance baselines would further support the usefulness of FBG. The rebuttal provided some additional results and clarifications about derivations and metrics. This reviewer was not fully convinced by the rebuttal.

Reviewer WucL noted that the proposed method is based on a principled theoretical framework (in contrast to heuristic methods like CFG or LIG), and that it outperforms CFG and performs competitively with LIG. The reviewer also noted that the paper includes results for class-conditional generation on ImageNet and text-to-image generation that demonstrate the applicability of FBG, and that it shows that FBG can be combined with CFG or LIG.

Reviewer WucL raised concerns regarding the computational overhead, posterior estimation bias, and scale of experiments. The rebuttal addressed some of these concerns.

Overall, this paper is interesting, novel, and proposes a theoretically grounded guidance method that outperforms CFG and is competitive with LIG (while having more theoretical foundation than LIG). The main drawback is the lack of complete experiments using larger-scale models. But even in the present form, it would be useful for the community.